# Towards Personalized Deep Research: Benchmarks and Evaluations

**Yuan Liang**$^{\diamond\spadesuit*}$, **Jiaxian Li**$^{\diamond*}$, **Yuqing Wang**$^{\diamond}$, **Piaohong Wang**$^{\diamond}$, **Motong Tian**$^{\diamond}$,
**Pai Liu**$^{\diamond}$, **Shuofei Qiao**$^{\spadesuit}$, **Runnan Fang**$^{\spadesuit}$, **He Zhu**$^{\diamond}$, **Ge Zhang**$^{\clubsuit}$, **Minghao Liu**$^{\triangle}$,
**Yuchen Eleanor Jiang**$^{\diamond}$, **Ningyu Zhang**$^{\spadesuit\dagger}$, **Wangchunshu Zhou**$^{\diamond\dagger}$

$^{\diamond}$OPPO   $^{\spadesuit}$Zhejiang University   $^{\clubsuit}$M-A-P   $^{\triangle}$2077.AI

`{liang_yuan,zhangningyu}@zju.edu.cn,zhouwangchunshu@oppo.com`

## Abstract

Deep Research Agents (DRAs) can autonomously conduct complex investigations and generate comprehensive reports, demonstrating strong real-world potential. However, existing evaluations mostly rely on close-ended benchmarks, while open-ended deep research benchmarks remain scarce and typically neglect personalized scenarios. To bridge this gap, we introduce **Personalized Deep Research Bench** (PDR-Bench), the first benchmark for evaluating personalization in DRAs. It pairs 50 diverse research tasks across 10 domains with 25 authentic user profiles that combine structured persona attributes with dynamic real-world contexts, yielding 250 realistic user-task queries. To assess system performance, we propose the PQR Evaluation Framework, which jointly measures Personalization Alignment, Content Quality, and Factual Reliability. Our experiments on a range of systems highlight current capabilities and limitations in handling personalized deep research. This work establishes a rigorous foundation for developing and evaluating the next generation of truly personalized AI research assistants[1].

## 1 Introduction

Recent advances in large language models (LLMs) have enabled the development of AI agents capable of conducting complex deep research. Early LLMs focus on isolated tasks like QA and translation, later advancing with tool integration for autonomous information retrieval and synthesis. More recently, a new class of advanced systems has emerged, known as Deep Research Agents (DRAs), including industry solutions (OpenAI, 2025b; Google DeepMind, 2025; xAI Team, 2025; Perplexity Team, 2025; Moonshot AI, 2025; ByteDance, 2025b) and open-source systems (Li et al., 2025b;c; Zhu et al., 2025a; Zhou et al., 2023; 2024; Wang et al., 2025a; Hu et al., 2025; Manus AI, 2025; MiroMind AI Team, 2025; ByteDance, 2025a; Li et al., 2025a; Tang et al., 2025; Shi et al., 2025; Zhu et al., 2025b). DRAs extend LLMs by incorporating dynamic reasoning, adaptive planning, and iterative tool use to acquire, aggregate, and analyze external information (Huang et al., 2025), thereby enabling end-to-end research workflows and the production of structured, comprehensive reports.

Despite these advances, to fully realize the potential of these intelligent systems in everyday human contexts, they must be able to adapt their behaviors and interactions to the specific needs of different users (Fischer, 2001; Kirk et al., 2024; Rafieian & Yoganarasimhan, 2023), a quality known as personalization. Important real-world decisions, from choosing a vehicle to making an investment, are strongly influenced by a user's unique needs, preferences, budget, and prior knowledge. In these scenarios, the agent's value lies not only in generating a comprehensive report, but also in acting as a personalized assistant that tailors its information filtering, reasoning, and recommendations. However, this critical dimension of personalization is a major blind spot for current evaluation methodologies.

---

\*   Equal Contribution.
$^{\dagger}$   Corresponding Author.
[1]   Code: https://github.com/OPPO-PersonalAI/PersonalizedDeepResearchBench.

Existing deep research benchmarks, including close-ended suites like GAIA, BrowseComp, HLE, and X-Bench (Mialon et al., 2023; Wei et al., 2025; Phan et al., 2025; Chen et al., 2025a) and open-ended ones like DeepResearch Bench, ResearcherBench, and DeepResearchGym (Du et al., 2025; Xu et al., 2025; Coelho et al., 2025), focus exclusively on factual accuracy and comprehensiveness, failing to assess user-specific adaptation. Conversely, existing personalization benchmarks such as LaMP, PersonaGym, PersonaLens and PersonaFeedback (Salemi et al., 2024; Samuel et al., 2025; Zhao et al., 2025; Tao et al., 2025) are confined to narrow domains like dialogue or recommendation and do not address the complex deep research. To the best of our knowledge, our work is the first to systematically incorporate personalization into the evaluation of DRAs, filling a critical gap in current research.

To address this gap, we introduce *Personalized Deep Research Bench*, a novel benchmark specifically designed to evaluate personalization in deep research agents. Our benchmark provides a rigorous framework for assessing how well agents can integrate user profiles into their research workflows, and whether their outputs are not only comprehensive and accurate, but also tailored and practically useful for the end user. By formalizing and evaluating this missing dimension, our work paves the way for the development of more effective and genuinely personal AI assistants.

Our main contributions are summarized as follows:

- We formally introduce the task of *personalized deep research*, which extends beyond generic information synthesis by requiring DRAs to adapt retrieval, reasoning and reporting to user personas.

- We propose *Personalized Deep Research Bench*, the first benchmark specifically targeting personalization in DRAs. It consists of 50 diverse tasks that span 10 domains and are paired with 25 real-world user profiles, yielding 250 unique user-task pairs, enabling systematic evaluation of both task complexity and persona-driven adaptation.

- We develop the *PQR Evaluation Framework*, a novel and comprehensive methodology that evaluates generated reports along three orthogonal dimensions: (P) *Personalization Alignment*, (Q) *Content Quality*, and (R) *Factual Reliability*, providing a holistic measure of agent utility in real-world research scenarios.

- We conduct extensive experiments across a broad spectrum of open-source DRAs, commercial deep research systems, LLMs with search tools and advancing memory systems, revealing both strengths and limitations in handling personalization.

## 2 RELATED WORK

### 2.1 EVALUATING DEEP RESEARCH CAPABILITIES

Evaluating DRAs requires benchmarks that go beyond traditional QA tasks to assess multi-turn retrieval, tool use, and structured report generation. Close-ended benchmarks such as GAIA, BrowseComp, HLE, and X-Bench (Mialon et al., 2023; Wei et al., 2025; Chen et al., 2025a) offer controlled evaluations, yet rely on synthetic tasks and fall short of reflecting the challenges of authentic research scenarios. Recently, open-ended deep research benchmarks have been proposed to specifically evaluate deep research capabilities. DeepResearch Bench (Du et al., 2025) offers 100 PhD-level tasks across 22 fields, introducing the RACE and FACT frameworks for report quality and retrieval assessment. Mind2Web 2 (Gou et al., 2025) features 130 real-world tasks with live web browsing and proposes the Agent-as-a-Judge framework for automated correctness and attribution. ResearcherBench (Xu et al., 2025) focuses on 65 frontier AI questions across 35 subjects with a dual rubric–factual evaluation. Additionally, BrowseComp-Plus (Chen et al., 2025b) extends BrowseComp (Wei et al., 2025) by pairing each query with curated documents and challenging negatives to isolate retriever and LLM contributions. DeepResearchGym (Coelho et al., 2025) provides an open-source sandbox with reproducible search APIs and standardized protocols for transparent, low-cost benchmarking. Nevertheless, these benchmarks focus on general research capabilities and lack metrics for personalization—the alignment of research with user-specific goals and preferences.

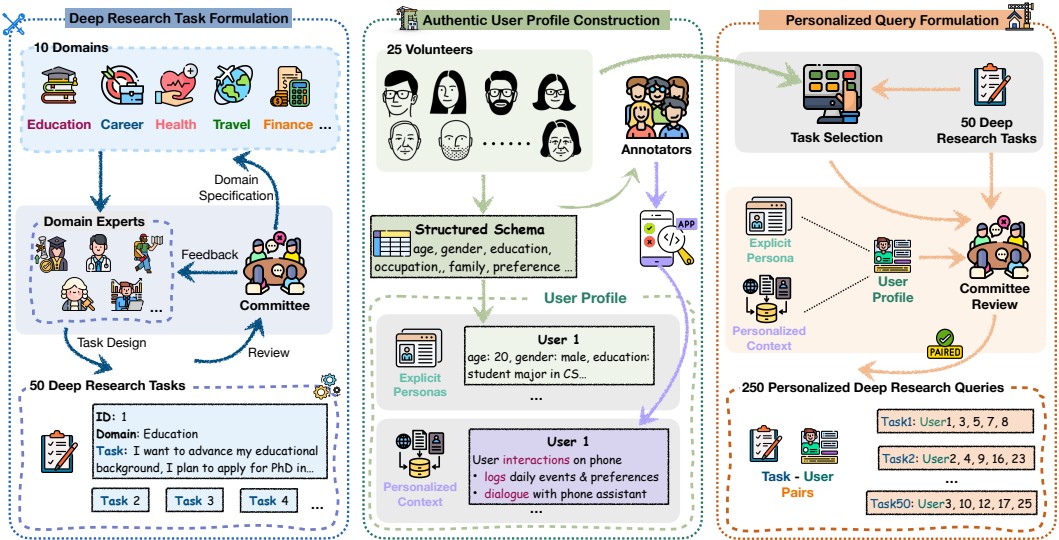

Figure 1: Benchmark Construction Pipeline: (1) Design 50 deep research tasks across 10 domains; (2) Build authentic user profiles from 25 volunteers; (3) Generate 250 personalized task-user pairs.

## 2.2 BENCHMARKING PERSONALIZATION PERFORMANCE

Meanwhile, most personalization benchmarks focus on general tasks and remain insufficient for complex deep research scenarios. LaMP (Salemi et al., 2024) introduces seven classification and generation tasks to evaluate the personalized output capacity of LLMs. PersonaGym (Samuel et al., 2025) introduces PersonaScore to evaluate the adherence of LLM agents to assigned personas at scale. PersonalLLM (Zollo et al., 2025) uses reward models to act as different user personas to evaluate response preference. AI Persona (Wang et al., 2024b) concentrates on the lifelong learning of user profiles with LLM-as-a-judge evaluation. Additionally, PersonaMem (Jiang et al., 2025) benchmarks the adaptability of LLMs to evolving user personas. PersonaFeedback (Tao et al., 2025) provides a large human-annotated benchmark for response tailoring to explicit personas. Person-aLens (Zhao et al., 2025) introduces LLM-based user and judge agents to assess personalization and task success in realistic dialogues.

Overall, current benchmarks either neglect personalization or fail to capture the complex nature of deep research, highlighting the pressing need for a new benchmark specifically designed to measure the personalized performance of DRAs.

# 3 BENCHMARK CONSTRUCTION

To rigorously evaluate the personalized capabilities of deep research agents, we introduce the Personalized Deep Research Bench, a benchmark designed to mirror the real-world personalized deep research scenarios. Its construction is grounded in two key components: a diverse set of deep research tasks and a collection of authentic, multifaceted user profiles, as shown in Figure 1.

## 3.1 DEEP RESEARCH TASK FORMULATION

**Domain Specification and Task Generation.** To begin, we defined a set of 10 distinct domains, $\mathcal{D} = \{d_1, d_2, ..., d_{10}\}$ covering major and impactful aspects of daily life (e.g., Career Development, Education, Healthcare, Financial Planning). To ensure that the tasks within each domain are both realistic and practically relevant, we collaborated with a diverse group of domain experts, such as travel bloggers, financial advisors and educational consultants, to design the initial set of tasks.

**Committee Review and Validation.** Each task underwent multistage validation by a committee of Master's/PhD researchers, data scientists, and product managers, following three principles: Complexity (↑): requiring multi-step reasoning, retrieval, and analysis; Clarity (↑): unambiguous

descriptions with clear objectives; Alignment ($\uparrow$): supporting the scenarios of personalized deep research.

Finally, we systematically formulated 5 balanced tasks per domain, yielding 50 tasks in total: $\mathcal{T} = t_i \mid i = 1, \ldots, 50$, with $t_i = (q_i, d(t_i))$ where $q_i$ is the query and $d(t_i) \in \mathcal{D}$ the domain. A parallel English set $\mathcal{T}_{EN}$ was also created, semantically aligned with the Chinese tasks.

## 3.2 AUTHENTIC USER PROFILE CONSTRUCTION.

A key innovation of our benchmark lies in the careful design of highly realistic and richly detailed authentic user profiles. We moved beyond synthetic or stereotyped characterizations by grounding our profiles in real user data.

**Structured Explicit Persona Collection.** We recruited 25 volunteers with diverse demographic profiles across age, profession, income, and life stage. After receiving standardized training on data authenticity and privacy, volunteers mapped their authentic personal details onto a specially designed persona schema, $\mathcal{S}$, which can be found in the Appendix D. This process yielded a set of 25 structured explicit ground-truth personas $\mathcal{P}_s$, denoted as: $\mathcal{P}_s = \{Ps_j \mid j = 1, \ldots, 25\}$.

**Dynamic Personalized Context Integration.** To complement these explicit personas with dynamic context, we employed professional annotators simulate the daily interactions of these collected personas through a phone APP. Over a period, they were instructed to: 1) Record naturalistic memory snippets ($m_j$), such as travel aspirations, health goals, and family plans; and 2) Conduct conversational interactions ($c_j$) with the intelligent assistant integrated in the app. This longitudinal data captures each user's evolving interests, habits, and implicit preferences. These multi-modal data streams were then processed by the built-in management system of the APP, $f_\theta$, to generate dynamic personalized contexts: $\mathcal{P}_c = \{Pc_j \mid Pc_j = f_\theta(m_j, c_j), j = 1, \ldots, 25\}$. Annotation details are in Appendix I.

For convenience, we define the complete user profile set $\mathcal{P}$ as the collection of paired structured explicit personas and dynamic personalized contexts:
$$\mathcal{P} = \{(Ps_j, Pc_j) \mid j = 1, \ldots, 25\}$$

## 3.3 PERSONALIZED DEEP RESEARCH QUERY FORMULATION

The final stage of benchmark construction involved the principled pairing of user profiles with deep research tasks to generate meaningful, personalized queries. We recognized that a random pairing would fail to capture the intrinsic relevance between a user and their research needs.

To address this, we employed a user-driven, committee-guided alignment protocol. Each of the 25 volunteers first reviewed the full task pool $\mathcal{T}$ and selected tasks that were personally relevant. Then, the committee curated and refined these selections through rigorous discussions, ensuring: (1) diversity of user profiles associated with each task, and (2) overall alignment between each user–task pair. This process yielded a user subset $\mathcal{P}_i \subset \mathcal{P}$ for each task $t_i$, where $|\mathcal{P}_i| = 5$.

Finally, a total of 250 personalized personalized deep research queries were formed:
$$\mathcal{Q} = \{(p, t_i) \mid i = 1, \ldots, 50, \ p \in \mathcal{P}_i\}, \quad |\mathcal{Q}| = 250$$
where each query combines one high-quality deep research task $t_i$ with a corresponding user profile $p$ from its assigned user set.

This benchmark faithfully mirrors real-world personalized deep research scenarios while providing a standardized, reproducible, and scalable evaluation setting. By jointly modeling task complexity, authentic user profile diversity, and motivational alignment, it provides a rigorous testbed for evaluating whether agents can effectively integrate user profiles into deep research and deliver truly personalized high-quality outputs.

## 4 EVALUATION METHODOLOGY

How do end-users judge the value of a Deep Research report? "Is this report for me?" (Personalization), "Is it well-crafted?" (Quality), and "Is the information true?" (Reliability). Existing eval-

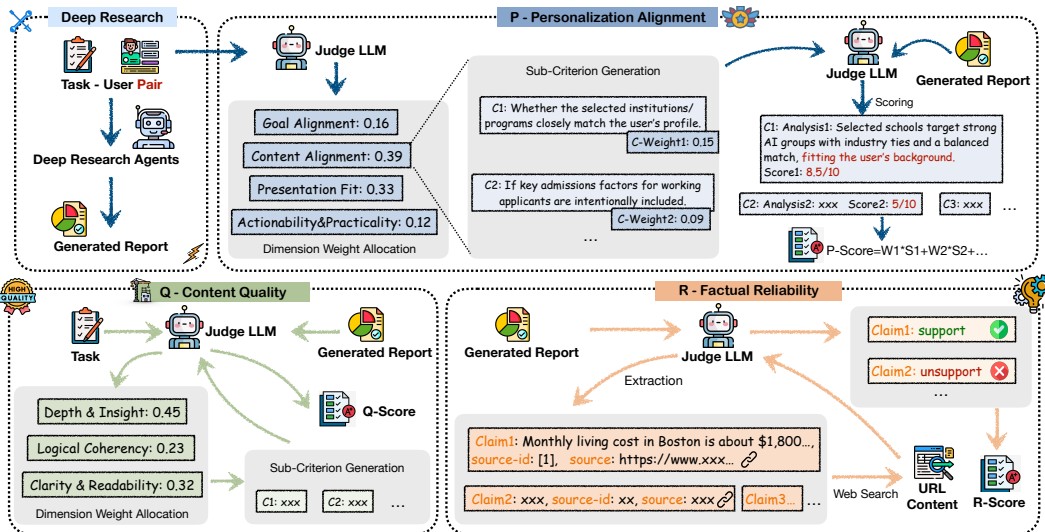

Figure 2: Overview of the PQR Evaluation Framework — A Multi-Dimensional Assessment System, Integrating Personalization Alignment (P), Content Quality (Q), and Factual Reliability (R).

uations, however, typically stress report quality or factual correctness, neglecting personalization. To systematically address these core concerns, we propose the **PQR Evaluation Framework**, a novel and comprehensive methodology assessing reports along three complementary axes: (P) Personalization Alignment, (Q) Content Quality, and (R) Factual Reliability. This joint consideration provides a holistic, user-centered assessment of Personalized Deep Research, as shown in Figure 2.

## 4.1 P - PERSONALIZATION ALIGNMENT

Evaluating the personalization of generated report is a significant challenge due to its subjective, multi-dimensional nature. Recent studies (Wang et al., 2024a; Guan et al., 2025; Zhu et al., 2025c) consistently emphasize that personalization evaluation should move beyond global correctness toward preference-aware and user-centered assessment paradigms, highlighting the need for individualized criteria rather than generic evaluation frameworks. Motivated by these insights and our requirement analysis, we introduce Personalization Alignment (P-Score), a dynamic evaluation framework that generates customized criteria and scores for each user–task pair.

The framework is built on four fundamental dimensions: **Goal Alignment (GOAL)**, **Content Alignment (CONT)**, **Presentation Fit (PRES)** and **Actionability & Practicality (ACTI)**. Detailed definitions of dimensions are provided in the Appendix E. The P-Score is computed via a three-stage, LLM-driven pipeline that operationalizes these dimensions into a quantitative score:

**Stage 1: Dynamic Dimension Weight Allocation.** An LLM, acting as a meta-evaluator, analyzes the input *task* $\mathcal{T}$ and *user persona* $\mathcal{P}_s$ to determine the relative importance of the four dimensions. This stage outputs a weight vector $W = \{w_d\}_{d \in D_P}$ for the set of personalization dimensions $D_P = \{\text{Goal Alignment, Content Alignment, ...}\}$, where $\sum_{d \in D_P} w_d = 1.0$.

**Stage 2: Granular Sub-Criterion Generation.** For each dimension $d \in D_P$, the LLM generates a set of granular sub-criteria $C_d^P = \{c_1, \ldots, c_n\}$, again conditioned on the *task* $\mathcal{T}$ and *user persona* $\mathcal{P}_s$. Each sub-criterion $c_i$ is assigned a weight $w_{c_i}$ such that $\sum_{i=1}^{n} w_{c_i} = 1.0$.

**Stage 3: LLM-Powered Scoring.** A separate LLM then scores the target report against the criteria. Given the report, $\mathcal{T}$, and $\mathcal{P}_s$, it assigns a score $s_{c_i} \in [0, 10]$ and justification for each sub-criterion $c_i \in C_d^P$.

The final P-Score $S_P$ is calculated by first computing each dimension score $S_d$ as the weighted average of its sub-criteria. Then, $S_P$ is obtained as the weighted average of the four dimension

scores using dynamically generated weights. This can be formally expressed as:

$$S_P = \sum_{d \in D_P} w_d \cdot S_d = \sum_{d \in D_P} w_d \left( \sum_{c_i \in C_d^P} w_{c_i} \cdot s_{c_i} \right) \tag{1}$$

where $w_d$ is the dimension weight, $w_{c_i}$ is the sub-criterion weight, and $s_{c_i}$ is the sub-criterion score.

## 4.2  Q – QUALITY OF CONTENT

Beyond personalization, we also assess the intrinsic quality of the generated report—its depth, insight, logic, clarity, and readability, regardless of user profiles. Quality is evaluated with respect to the task $\mathcal{T}$ and standards of rigorous research writing.

We define three dimensions: **Depth & Insight (DEIN)**, **Logical Coherence (LOGC)**, and **Clarity & Readability (CLAR)** (see Appendix E).

**Evaluation Process.**  Following the dynamic criterion principle, an LLM meta-evaluator (i) assigns weights $\{w_d\}_{d \in D_Q}$ to the three dimensions, and (ii) generates a set of task-specific sub-criteria $C_d^Q$ for each dimension. A separate LLM scorer then rates the report against this criterion, producing a score $s_{c_i} \in [0, 10]$ with justification for each sub-criterion $c_i \in C_d^Q$. The final Q-Score is a hierarchical weighted average:

$$S_Q = \sum_{d \in D_Q} w_d \cdot S_d = \sum_{d \in D_Q} w_d \left( \sum_{c_i \in C_d^Q} w_{c_i} \cdot s_{c_i} \right) \tag{2}$$

where $w_d$ is the dimension weight, $w_{c_i}$ is the sub-criterion weight, and $s_{c_i}$ is the sub-criterion score.

## 4.3  R – FACTUAL RELIABILITY.

Although factuality metrics (Min et al., 2023; Wei et al., 2024; Chern et al., 2023) exist, they are primarily designed to verify atomic facts against knowledge bases or external search results. They remain unsuitable for deep research setting, where factuality must be evaluated through retrieved citations to assess both the factual reliability of the report and the agent's capacity in utilizing web information. We therefore assess report reliability via an automated factual grounding framework, inspired by ResearcherBench (Xu et al., 2025) and DeepResearch Bench (Du et al., 2025). The process has three stages:

**Claim Extraction and Deduplication.**  A Judge LLM is employed to extract all verifiable factual claims with their sources, forming a set of triplets $\mathcal{TRI} = \{(c_i, \mathrm{idx}_i, \mathrm{source}_i)\}_{i=1}^N$, where uncited claims have empty sources (see Appendix J.3). A second pass deduplicates claims:

$$\mathcal{TRI}\mathrm{unique} = \mathrm{Deduplicate}(\mathcal{TRI}), \tag{3}$$

yielding $N_{\mathrm{total}}$ unique claims, of which $N_{\mathrm{cited}}$ are cited.

**Automated Verification.**  For each unique triplet $(c_i, \mathrm{idx}_i, \mathrm{source}_i) \in \mathcal{TRI}_{\mathrm{unique}}$, we use the Jina Reader API to retrieve the source content $Content_i$. Then the Judge LLM checks support:

$$v_i = \begin{cases} 1, & \text{if } c_i \text{ is } supported \text{ by } Content_i, \\ 0, & \text{if } c_i \text{ is } unsupported \text{ or } unknown. \end{cases} \tag{4}$$

**Metric Calculation.**  We compute two key metrics from the verification results:

- **Factual Accuracy (FA)** Measures the reliability of provided citations. It is the percentage of claims that are factually verified and supported by their corresponding source material.
- **Citation Coverage (CC)** Assesses the proportion of factual claims in a report that are supported by explicit citations, reflecting how well the content is evidence-based.

Finally, we average FA and CC to derive a single Factual Reliability score, $S_R$:

$$\mathrm{FA} = \frac{\sum_{i=1}^{N_{\mathrm{cited}}} v_i}{N_{\mathrm{cited}}} \times 10, \quad \mathrm{CC} = \frac{N_{\mathrm{cited}}}{N_{\mathrm{total}}} \times 10, \quad S_R = \frac{\mathrm{FA} + \mathrm{CC}}{2}. \tag{5}$$

## 4.4 FINAL SCORE AGGREGATION

To obtain a holistic measure of the report, we define the final overall score as a arithmetic mean over the three dimension scores:

$$S_{\text{overall}} = \frac{S_P + S_Q + S_R}{3} \tag{6}$$

where $S_P$, $S_Q$, and $S_R$ denote the scores for personalization, quality and factual reliability respectively. This aggregation provides a straightforward and comprehensive measure of the personalized deep research report.

## 5 EXPERIMENTS

### 5.1 EXPERIMENTAL SETTINGS

We benchmarked a diverse set of systems, including commercial deep research systems: Gemini-2.5-Pro Deep Research, O3 Deep Research, Perplexity Deep Research (Google DeepMind, 2025; OpenAI, 2025b; Perplexity Team, 2025), open-source deep research agents: Deerflow, Oagents, Miroflow (ByteDance, 2025a; Zhu et al., 2025a; MiroMind AI Team, 2025), and leading LLMs with search tools: Gemini-2.5-Pro-Search, Claude-3.7-Sonnet-Search, Perplexity-Sonar-Reasoning-Pro, GPT-4.1-Search-Preview (DeepMind; Anthropic; AI, 2025; OpenAI, a). Due to computational constraints, the evaluation was performed on a subset of 150 representative queries. GPT-5 (OpenAI, 2025a) was utilized as the judge model for Personalization (P) and Quality (Q) metrics, while the more efficient GPT-5-Mini (OpenAI, b) served as the judge for the Reliability (R) metric, ensuring a balance of advanced reasoning and efficiency (more details in Appendix B and F).

### 5.2 MAIN RESULTS

Table 1: Evaluation results of Personalized Deep Research Bench under the *Task w/Persona* configuration. The best results in each column are highlighted in **bold**, and the second-best results are underlined.

| Model | Overall | Personalization | | | | Quality | | | Reliability | |
|---|---|---|---|---|---|---|---|---|---|---|
| | | GOAL | CONT | PRES | ACTI | DEIN | LOGC | CLAR | FA | CC |
| *Commercial Deep Research Agents* | | | | | | | | | | |
| Gemini-2.5-Pro Deep Research | **6.58** | 5.27 | 5.78 | **5.83** | 4.56 | 5.32 | 6.13 | **6.16** | **8.40** | **9.26** |
| O3 Deep Research | 6.11 | **5.67** | **5.95** | 5.57 | **5.10** | **5.68** | **6.40** | 5.58 | 6.84 | 7.14 |
| Perplexity Deep Research | 5.99 | 4.69 | 4.93 | 4.72 | 4.33 | 4.93 | 5.43 | 4.68 | 7.68 | 9.02 |
| *Open-Source Deep Research Agents* | | | | | | | | | | |
| OAgents | **6.64** | **6.68** | 6.44 | **7.13** | **6.92** | **6.99** | **7.44** | **6.85** | 3.77 | **8.32** |
| DeerFlow | 5.30 | 5.20 | 4.97 | 6.71 | 5.41 | 5.43 | 6.25 | 6.44 | 6.85 | 2.32 |
| MiroFlow | 5.78 | 6.65 | **6.45** | 7.03 | 6.65 | 6.53 | 7.31 | 6.68 | **7.29** | 0.44 |
| *LLM with Search Tools* | | | | | | | | | | |
| Gemini-2.5-Pro w/Search | **5.53** | **4.85** | **5.20** | 5.61 | 4.19 | **4.54** | **5.57** | 5.41 | 6.99 | **6.62** |
| Claude-3.7-Sonnet w/Search | 4.83 | 4.27 | 4.24 | 5.43 | **4.28** | 4.26 | 5.09 | 5.34 | 8.27 | 2.37 |
| Perplexity-Sonar-Reasoning-Pro | 5.02 | 4.27 | 4.37 | 5.27 | 4.15 | 4.22 | 5.03 | 5.23 | **8.44** | 3.67 |
| GPT-4.1 w/Search | 4.28 | 4.59 | 4.86 | **5.74** | 4.07 | 4.21 | 5.27 | **5.54** | 6.75 | 0.10 |

The evaluation results on the Personalized Deep Research Bench under the Task w/Persona configuration (Task and Persona are explicitly provided to the agent) are shown in Table 1. Our analysis reveals several key findings regarding the performance of different model categories.

**Open-source Agents Excel in Personalization.** Open-source agents achieve the strongest personalization, with OAgents achieving the top score (6.64) and leading most sub-metrics, including GOAL (6.68), PRES (7.13), and LOGC (7.44). MiroFlow also performs competitively, outperforming OAgents in CONT (6.45) and FA (7.29). However, reliability remains their weakness: OAgents suffers from low factual accuracy (3.77), while both MiroFlow and DeerFlow show poor citation coverage.

**Commercial Agents Provide Balanced Quality and Reliability.** Commercial systems achieve slightly lower personalization but higher reliability and consistent quality. Gemini-2.5-Pro Deep Research leads this group (6.58), achieving top FA (8.40), CC (9.26), and solid quality scores (DEIN:

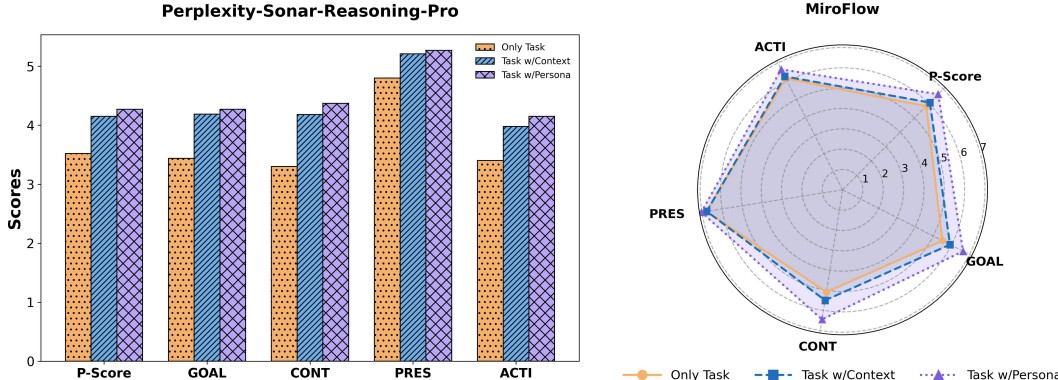

Figure 3: Analysis on Personalization Metrics for Sonar-Reasoning-Pro and MiroFlow

5.32, LOGC: 6.13, CLAR: 6.16). O3 Deep Research follows closely (6.11), leading in personalization within this category (GOAL: 5.67, CONT: 5.95) and maintaining competitive quality (DEIN: 5.68, LOGC: 6.40, CLAR: 5.58). In summary, commercial agents are reliable and robust in quality, but they lag moderately behind open-source agents in personalization.

**LLMs with Search Tools Fall Short.** Search Tools equipped LLMs underperform specialized agents. Gemini-2.5-Pro w/Search is the strongest in this group (5.53), while others, such as Perplexity-Sonar-Reasoning-Pro, achieve high FA (8.44) but poor CC and weak personalization. GPT-4.1 w/Search, for example, nearly fails in CC (0.10). These results indicate that adding search alone is insufficient to reach the personalization and quality of dedicated deep research agents.

## 5.3 Impact of Information Availability on Personalization

While the previous section evaluated agents with explicit personas, a more realistic scenario involves inferring user needs from conversational or interaction context, as explicit personas are rarely available. To examine this, we conducted a comparative experiment under three conditions: Task Only (the agent receives only the task), Task w/Context (the task plus user's conversational or interaction context), and Task w/Persona (the task plus an explicit user persona, consistent with our main experiment). We present the results in Table 2 and Figure 3. A more detailed results are in Appendix G.

Table 2: Evaluation results on the Personalization across different personalization settings. The table shows results for three configurations: *Task Only*, *Task w/Context*, and *Task w/Persona*. Best scores in each column are highlighted in **bold**, second-best in underlined.

| Model | Setting | Personalization | | | | |
|---|---|---|---|---|---|---|
| | | P-Score | GOAL | CONT | PRES | ACTI |
| OAgents | *Task Only* | 6.17 | 5.91 | 5.42 | 6.90 | 6.51 |
| | *Task w/Context* | 6.53 | 6.32 | 5.99 | 7.04 | 6.81 |
| | *Task w/Persona* | **6.78** | **6.68** | **6.44** | **7.13** | **6.92** |
| O3 Deep Research | *Task Only* | 5.13 | 5.14 | 5.08 | 5.62 | 5.03 |
| | *Task w/Context* | **5.48** | 5.58 | 5.67 | **5.70** | **5.29** |
| | *Task w/Persona* | 5.46 | **5.67** | **5.95** | 5.57 | 5.10 |
| Gemini-2.5-Pro w/Search | *Task Only* | 3.96 | 3.91 | 3.86 | 5.53 | 3.70 |
| | *Task w/Context* | 4.55 | 4.66 | 4.95 | 5.59 | 4.09 |
| | *Task w/Persona* | **4.70** | **4.85** | **5.20** | **5.61** | **4.19** |

**More Information Consistently Yields Better Personalization.** Across all systems, personalization scores (P-Score) increase with the user information (persona or context) provided. This trend holds for all sub-metrics, confirming the intuitive hypothesis that access to user-specific data is crucial for tailoring research outputs.

**Explicit Personas Outperform Context.** Context improves performance over the baseline, but the largest gains come from explicit personas. For instance, OAgents' GOAL score increases from 6.32 (Context) to 6.68 (Persona), a larger jump than the improvement from Task Only to Context.

This indicates that while agents can partially leverage implicit context, they struggle to fully extract user preferences from unstructured, implicit data. Explicit personas, by contrast, provide a stronger and more accessible personalization signal.

## 5.4 Boosting Personalization via Context-Aware Memory Systems

The preceding analysis reveals that while contextual information is beneficial, agents struggle to distill it into an actionable user understanding as effectively as when provided with an explicit persona.

To address this, we designed a second experiment to test whether advanced memory systems can transform unstructured *context* into explicit *persona* to perform personalized deep research. we evaluated 50 suitable queries on three systems: Mem0 (Chhikara et al., 2025), Memory OS (Kang et al., 2025), and O-Mem (Wang et al., 2025b), on their ability to extract, integrate, and infer user preferences from *context* to drive downstream deep research systems. Results indicate potential for improving higher-level reasoning and user information integration.

Table 3: Evaluation results on for different memory systems under the *Task w/Context* setting, using Perplexity Deep Research. Currently, most memory systems can only align content with user characteristics, so we prioritize GOAL and CONT scores. We also display other metrics for clarity. The best metric is highlighted in **bold**. Underlined denotes the second highest.

| Method | Personalization | | | | |
|---|---|---|---|---|---|
| | P-Score | GOAL | CONT | PRES | ACTI |
| No Memory | 3.69 | 3.88 | 3.74 | 3.90 | 3.46 |
| Mem0 | 3.55 | 3.73 | 3.55 | 3.77 | 3.36 |
| Memory OS | 3.88 | 4.06 | 3.97 | 4.09 | 3.66 |
| O-Mem | **4.26** | **4.47** | **4.43** | **4.34** | **4.00** |
| Task w/Persona | 4.58 | 4.69 | 4.93 | 4.72 | 4.33 |

As shown in Table 3, memory systems yield varied but promising results. O-Mem outperforms both *No Memory* baseline and other systems, while Mem0 underperforms. However, a significant gap remains between the best system and ideal *Task w/Persona* performance, indicating that current memory systems struggle to fully synthesize information from context. This gap highlights the need for future research on memory systems that combine factual retrieval with higher-level reasoning and abstraction, moving beyond storage toward constructing dynamic, persona-like models of users.

## 5.5 Alignment with Human Consistency

To validate the evaluation framework, we conducted a systematic study comparing the judgments of LLMs against human experts. We sampled 15 representative queries and generated responses from two deep research agents: MiroFlow and O3 Deep Research. A panel of human evaluators scored these reports using the same criteria, establishing a ground truth for our comparison.

We designed two complementary metrics to quantify this alignment: Pairwise Comparison Agreement (PCA) measures the percentage of the LLM judge and human experts agree on which of the two reports is better for a specific criterion. Mean Absolute Rating Deviation (MARD) measures the average absolute difference between the scores assigned by the LLM and human judges. Detailed mathematical formulations for these metrics can be found in Appendix C.

Based on the results shown in Table 4, GPT-5 achieved the highest PCA and lowest MARD, indicating the strongest agreement with human judgments, while maintaining a reasonable cost ($0.68 per query). We finally select `GPT-5` as our primary judge model.

Table 4: Alignment results of judge LLMs with human ratings. PCA is reported as proportion of agreement (higher is better), MARD as mean absolute deviation (lower is better), Avg. Cost is measured in US dollars ($). The best metric is highlighted in **bold**.

| Judge LLM | PCA ↑ | MARD ↓ | Avg. Cost ($) ↓ |
|---|---|---|---|
| GPT-5 | **0.43** | **1.40** | 0.68 |
| Claude-3.7-Sonnet | 0.39 | 1.44 | 0.97 |
| Gemini-2.5-Pro | 0.40 | 2.33 | **0.61** |

## 6 CONCLUSION

In conclusion, our work addresses the critical gap in DRAs evaluation by introducing the Personalized Deep Research Bench, the first benchmark of its kind featuring 250 realistic queries that pair 50 diverse deep research tasks across 10 domains with 25 authentic user profiles. Along with the PQR Evaluation Framework, which jointly measures personalization, content quality, and factual reliability, our study reveals both the potential and current limitations towards personalized deep research. By establishing this rigorous foundation, our work paves the way for developing and benchmarking the next generation of truly personalized and effective AI research assistants.

## 7 ETHICS STATEMENT

This work strictly adheres to the ICLR Code of Ethics. The study protocol underwent internal compliance, privacy, safety review and obtained ethical approval within our institution. The research was conducted in accordance with applicable institutional and national research ethics regulations. User profiles were collected from volunteers under informed consent. Participants were informed of the research purpose, data usage, anonymization procedures, and public release format, and could withdraw at any time without consequence. All data were strictly anonymized prior to release. The publicly released benchmark contains only de-identified and abstracted derivatives of the original profiles; no personally identifiable information (PII) is included. Annotators and human evaluators participated voluntarily under informed consent and were fairly compensated. They were instructed to ensure accuracy, neutrality, and confidentiality in their annotations and evaluations. To promote fairness and mitigate potential biases, tasks and profiles were designed to cover diverse domains, age ranges, professions, income levels, and life stages. The study does not involve sensitive or harmful content and was conducted in a controlled and ethically responsible manner.

## 8 REPRODUCIBILITY STATEMENT

To ensure reproducibility, we provide comprehensive details across the paper and appendix. Section 3 and Appendix I describes benchmark construction, including task design, user profiles collection, and query pairing. The evaluation framework is detailed in Section 4, including the dynamic weight allocation, granular criterion generation, and scoring methodology for each dimension. Section 5 and Appendix B outline experimental setups, systems, and configurations. Prompt templates (Appendix J), persona schema (Appendix D), and evaluation dimensions (Appendix E) are fully documented. We have submitted our evaluation data in the Supplementary Material. Due to Open-Review's file size limit, we only upload a subset of them. We will fully release the benchmark immediately after the double blind review process. We invite the community to build upon this work in advancing personalized deep research agents.

### ACKNOWLEDGMENTS

We would like to express our sincere gratitude to the anonymous reviewers for their thoughtful and constructive feedback. This work was supported by the National Natural Science Foundation of China (No. 62576307, No. NSFCU23B2055, No. NSFCU19B2027), the Fundamental Research Funds for the Central Universities (226-2023-00138), Yongjiang Talent Introduction Programme (2021A-156-G), and Information Technology Center and State Key Lab of CAD&CG, Zhejiang University.

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

## A    LIMITATIONS

This work still has some limitations that must be admited: *a)* The collection of user personas and context annotations was conducted in Chinese. Although a parallel English version was provided, the underlying content remains linguistically and culturally constrained. *b)* Due to computational constraints, our main experiments were conducted on a selected subset of queries rather than the full benchmark. For future work, we plan to expand persona construction and context annotation beyond the Chinese-centric setting, aiming for more diverse and cross-lingual coverage. We also intend to scale up our experimental scope to fully utilize all benchmark queries and explore richer evaluation protocols under varied computational settings.

## B    EXPERIMENT DETAIL

### B.1    CONFIGURATION OF METHODS

To ensure a fair comparison among different agents, we standardized their execution budgets and search configurations. Specifically, open-source deep research agents: Deerflow, Oagents, Miroflow are all standardized on GPT-5-Mini as the base LLM. The maximum number of execution steps was set to 8 for OAgent. For Deerflow: `max_step_num` and `max_plan_iterations` are default set to 3 and 1. To Miroflow: `max_turns` and `max_tool_calls_per_turn` in `main_agent` and `sub_agents` are both default set to 20 and 10. All agents relied on SerperAPI for web search and Jina for web content retrieval. For agents equipped with built-in web search tools, we set the `reasoning_effort` parameter to medium. In addition, for sonar-reasoning-pro, the `search_context_size` parameter was also set to medium.

### B.2    DATA SELECTION

Given the high cost of running our full evaluation pipeline, we first reduced the original set of 250 queries. These queries covered 50 distinct tasks, each originally paired with 5 personas. To make the evaluation more tractable, we limited each task to 3 representative personas, resulting in a reduced set of 150 queries. From this subset, we further selected 50 queries that reflect a broad range of personalization demands across different user goals and characteristics. These queries were chosen based on their potential to reveal how effectively a memory system can adapt its responses to individual users. Since current memory systems are primarily limited to aligning content with user profiles rather than deeper task-level adaptation, our evaluation emphasizes Goal Alignment and Content Alignment as the core metrics. Additional metrics are also reported for completeness and transparency.

## C    FORMULATION OF HUMAN CONSISTENCY METRICS

This section provides the detailed mathematical definitions for the metrics used to evaluate the alignment between LLM judges and human experts.

**Pairwise Comparison Agreement (PCA).**    For each query $q$ and criterion $c$, let $A$ and $B$ denote the two reports being compared. We denote the model's scores as $m_{q,c}^A$ and $m_{q,c}^B$, and the human ground truth scores as $h_{q,c}^A$ and $h_{q,c}^B$. PCA is defined as the proportion of cases where the model's preference order matches the humans' preference order:

$$\text{PCA} = \frac{1}{N} \sum_{q,c} \mathbf{1}\Big[\text{sgn}\big(m_{q,c}^A - m_{q,c}^B\big) = \text{sgn}\big(h_{q,c}^A - h_{q,c}^B\big)\Big],$$

where $N$ is the total number of query–criterion pairs, $\text{sgn}(x)$ is the sign function, and $\mathbf{1}[\cdot]$ is the indicator function, which equals 1 if the condition is true and 0 otherwise.

**Mean Absolute Rating Deviation (MARD).**    For each query $q$, report $r \in \{A, B\}$, and criterion $c$, the MARD is the overall mean of the absolute deviations between the model scores ($m_{q,c}^r$) and

the human scores ($h_{q,c}^r$). It is calculated as:

$$\text{MARD} = \frac{1}{\sum_q 2|C_q|} \sum_q \sum_{r \in \{A,B\}} \sum_{c \in C_q} |m_{q,c}^r - h_{q,c}^r|,$$

where $|C_q|$ is the number of criteria for query $q$, and the term $2|C_q|$ accounts for the two reports being evaluated for each criterion.

## D  PERSONA SCHEMA

Table 5: Predefined Schema For Persona Collection (All Private Data Anonymized for Release)

| **Predefined Persona Schema.** | |
| --- | --- |
| **Basic Attributes** | |
| Identity Characteristics | Name, Age, Gender, Occupation |
| Family Status | Family Members and Relationships, Pets |
| Long-term Spatial Characteristics | Permanent Residence, Hometown |
| **Behavioral Characteristics** | |
| Online Usage Habits | High-frequency Apps and Usage Duration, Online Social Behavior |
| Offline Long-term Behavior | Daily Routine, Consumption Cycle, Consumption Characteristics, Periodic Mobility |
| **Environment** | |
| Time | Time Preference |
| Geographical Location | Frequent Places, Travel Radius |
| **Personality Traits** | Personality, Decision-making Style, Shopping Preference |
| **Preferences and Interests** | |
| Lifestyle Preferences | Diet, Accommodation Preferences, Shopping, Services |
| Travel Preferences | Frequency, Destinations, Travel Style |
| Content Preferences | Article Collection, Short Videos, Screenshots, Books, Movies, Singers, Actors |
| Exercise Preferences | Exercise Habits, Exercise Goals, Exercise Types, Other Investments, Exercise Locations |
| **Health Status** | |
| Physical Condition | Medical History, Mental Condition, Physical Fitness |
| Health Needs | Health Needs |
| **Financial Information** | |
| Financial Status | Income Structure, Asset Status, Consumption Characteristics, Debt Situation |
| Investment Experience | Investment Background, Knowledge Level |
| Risk Management | Risk Appetite |

# E    DEFINITIONS OF EVALUATION DIMENSIONS

Table 6: Dimensions and Definitions For Personalization Evaluation.

| Dimension | Description |
|---|---|
| Goal Alignment | How well the report addresses the user's explicit and implicit goals. |
| Content Alignment | The suitability of the report's topic, depth, and breadth for the user's knowledge and interests. |
| Presentation Fit | The alignment of the report's language, structure, and style with the user's comprehension and preferences. |
| Actionability&Practicality | The extent to which the report offers practical value for decision-making or action. |

Table 7: Dimensions and Definitions For Quality Evaluation.

| Dimension | Description |
|---|---|
| Depth & Insight | The analytical richness, originality of thought, and critical perspective exhibited by the report. |
| Logical Coherence | The logic and coherence of the report's reasoning, ensuring ideas are rigorous and easy to follow. |
| Clarity & Readability | The report's language, information presentation, and formatting. |

# F    RUNNING COST

Table 8: Running Cost Comparison Across Models and Personalization Settings (per query)

| Model / Setting | Task Only | Task w/ Persona | Task w/ Context |
|---|---|---|---|
| *Open-Source DRAs (GPT-5-Mini based)* | | | |
| OAgents | $1.60 | $1.70 | $3.00 |
| DeerFlow | $0.50 | $0.57 | $1.20 |
| MiroFlow | $1.00 | $1.11 | $2.10 |
| *LLM with Search Tools* | | | |
| Gemini-2.5-Pro w/ Search | $0.05 | $0.06 | $0.24 |
| Claude-3.7-Sonnet w/ Search | $0.03 | $0.04 | $0.31 |
| Perplexity-Sonar-Reasoning-Pro | $0.02 | $0.03 | $0.78 |
| GPT-4.1 w/ Search | $0.02 | $0.02 | $0.31 |

# G  A More Detailed Experiments Results

Table 9: Evaluation results on the Personalization across different personalization settings to various agents. The table shows results for three configurations: *Task Only*, *Task w/Context*, and *Task w/Persona*. Best scores in each column are highlighted in **bold**, second-best in underlined.

| Model | Setting | P-Score | GOAL | CONT | PRES | ACTI |
|---|---|---|---|---|---|---|
| *Commercial DeepResearch Agents* | | | | | | |
| Gemini-2.5-Pro | *Task Only* | 4.57 | 4.12 | 3.90 | **6.49** | 4.77 |
| | *w/Context* | 4.70 | 4.84 | 5.17 | 5.66 | 4.18 |
| | *w/Persona* | **5.12** | **5.27** | **5.78** | 5.83 | **4.56** |
| O3 | *Task Only* | 5.13 | 5.14 | 5.08 | 5.62 | 5.03 |
| | *w/Context* | **5.48** | 5.58 | 5.67 | **5.70** | **5.29** |
| | *w/Persona* | 5.46 | **5.67** | **5.95** | 5.57 | 5.10 |
| Perplexity | *Task Only* | 3.58 | 3.47 | 3.38 | **4.82** | 3.47 |
| | *w/Context* | 4.19 | 4.23 | 4.29 | 4.56 | 4.06 |
| | *w/Persona* | **4.58** | **4.69** | **4.93** | 4.72 | **4.33** |
| *Open-Source DeepResearch Agents* | | | | | | |
| OAgents | *Task Only* | 6.17 | 5.91 | 5.42 | 6.90 | 6.51 |
| | *w/Context* | 6.53 | 6.32 | 5.99 | 7.04 | 6.81 |
| | *w/Persona* | **6.78** | **6.68** | **6.44** | **7.13** | **6.92** |
| Deerflow | *Task Only* | 5.11 | 4.77 | 4.41 | 6.67 | 5.31 |
| | *w/Context* | 5.02 | 4.77 | 4.47 | 6.60 | 5.09 |
| | *w/Persona* | **5.38** | **5.20** | **4.97** | **6.71** | **5.41** |
| Miroflow | *Task Only* | 5.82 | 5.51 | 5.09 | 6.78 | 6.15 |
| | *w/Context* | 6.07 | 5.93 | 5.50 | 6.76 | 6.26 |
| | *w/Persona* | **6.65** | **6.65** | **6.45** | **7.03** | **6.65** |
| *LLM with Search Tools* | | | | | | |
| Gemini-2.5-Pro | *Task Only* | 3.96 | 3.91 | 3.86 | 5.53 | 3.70 |
| | *w/Context* | 4.55 | 4.66 | 4.95 | 5.59 | 4.09 |
| | *w/Persona* | **4.70** | **4.85** | **5.20** | **5.61** | **4.19** |
| Claude-3.7-Sonnet | *Task Only* | 4.00 | 3.83 | 3.63 | 5.32 | 3.99 |
| | *w/Context* | 3.85 | 3.87 | 4.06 | 4.80 | 3.54 |
| | *w/Persona* | **4.37** | **4.27** | **4.24** | **5.43** | **4.28** |
| Sonar-Rea-Pro | *Task Only* | 3.52 | 3.44 | 3.30 | 4.80 | 3.40 |
| | *w/Context* | 4.15 | 4.19 | 4.18 | 5.21 | 3.98 |
| | *w/Persona* | **4.27** | **4.27** | **4.37** | **5.27** | **4.15** |
| GPT4.1 | *Task Only* | 3.79 | 3.71 | 3.63 | 5.44 | 3.55 |
| | *w/Context* | 4.41 | 4.43 | 4.52 | 5.70 | **4.08** |
| | *w/Persona* | **4.52** | **4.59** | **4.86** | **5.74** | 4.07 |

# H  CASE STUDY

**Deep Research Task**

I am a beginner runner preparing for my first full marathon in the next three months, with the goal of completing the race safely and without injury. I can train about five days a week for roughly one hour per session, mainly running in nearby parks or residential roads.

I do not have professional running equipment—only a pair of basic running shoes. I currently have no running experience and no specific pace targets, but I hope to gradually increase my mileage while considering my daily life and work schedule.

The training plan should be easy to follow, sustainable, and adaptable. Please create a personalized training plan based on these conditions

**Personal Description**

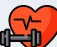 Male, 34-year-old, Supply chain management professional

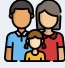 Minimal exercise engagement, High work and personal life demands

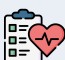 Married status, 6-year-old daughter, Alaskan Malamute ownership

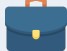 Mild obesity condition, Neck discomfort symptoms, Overall health improvement aspirations

High-demand professional role, Frequent business travel requirements

**Personal Description**

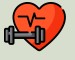 Female graduate student, Clinical psychology specialization

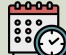 Three weekly exercise sessions,Yoga and light jogging preferences,Home-based and proximate outdoor exercise venues

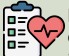 Morning chronotype orientation, Early morning study and writing optimization

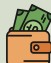 Mild obesity condition, Neck discomfort symptoms, Overall health improvement aspirations

Scholarship dependency, Part-time employment supplementary income

**Personalized Report**

1. Designate weekends as family fitness integration periods, incorporating Alaskan Malamute into running activities for joint family-dog exercise engagement.
2. Develop gradual weight reduction protocols with specialized running posture guidance to prevent neck discomfort exacerbation.
3. Implement business travel-adapted training solutions utilizing hotel fitness facilities with progressive beginner-oriented training progression plans.

**Personalized Report**

1. Integrate running as a mindfulness-based practice utilizing psychological training for enhanced self-observation and emotional regulation.
2. Optimize morning chronotype by establishing running as the primary morning routine for maximum cognitive and physical benefits.
3. Facilitate peer interaction through university or community running club participation and social media training documentation.
4. Implement cost-effective equipment selection with complimentary training resources and mobile application recommendations to minimize financial barriers.

Figure 4: Case Study of Personalized Deep Research

# I  DATA ANNOTATION

Our context dataset was manually annotated by 6 trained annotators, resulting in 5,939 labeled instances. The process required a total effort of 85 person-days and a budget of approximately $6,000 USD.

---

**Data Annotation Protocol**

**Task Objective**
The core of this task is to simulate the behavior of a specific user persona by collecting various types of information that this user would likely encounter, consume, or generate in their daily life. The final goal is to create a high-quality *memory* database for each persona that accurately reflects their unique characteristics.

**Core Quality Standard: Reversibility**
This is the most critical quality standard for this task. Every piece of data you collect must clearly point to a specific trait of the user persona.

- **Verification Method:** After collection, review all data and assess whether its content is sufficient to reverse-infer the user's key persona traits, such as their profession, interests, and personality.

- **Acceptance Criteria:** All collected data must pass the *reversibility* test to be considered high-quality. If the data is too generic and the persona cannot be inferred from it, the data will be deemed non-compliant.

**Collection Based on Persona Preferences**

- Carefully read every preference tag in the user persona description.

- For each preference, collect **at least five** pieces of related content.

**Ensure Diversity of Sources and Types**

- **Source Diversity:** Do not limit collection to a single platform. Data can be gathered from various apps (e.g., Twitter, Instagram, Reddit), websites, forums, etc.

- **Type Diversity:** Diversify the format of the data you collect. Examples include:
    - Screenshots of social media posts.
    - Screenshots of conversations with friends (that reflect opinions or preferences).
    - Links and titles of articles, news, or videos.
    - Screenshots of purchase histories or product reviews.

**Requirement for Conversational Content**
Between 20% and 50% of the total data collected for each user should be in a conversational format. This helps to more vividly showcase the user's personality and communication habits.

**Add Reasonable Noise for Authenticity**
The content you collect does not need to be an exact match to the persona's description. You can add relevant and reasonable details or *noise* to make the data appear more authentic, as if generated by a real user.

- **Example:** If the persona *likes basketball*, you could collect a news article about a recent Lakers game or a screenshot of a conversation with a friend debating whether Jordan or LeBron is the better player.

- **Note:** Any added *noise* must not contradict other defined attributes in the persona, such as spending habits, personality, or profession.

**Quantity vs. Quality**
There are no strict quantitative requirements for data collection. Please prioritize quality and collect as much rich data as possible. Quality always takes precedence over quantity.

**Deliverables and Annotation Requirements**

- **Deliverable:** For each user persona, export a separate `aimemory` database file.

- **Content Annotation:** During collection, each piece of content must be given a clear title and be correctly associated with its corresponding persona.

---

# J PROMPT TEMPLATES

## J.1 PROMPTS IN PERSONALIZATION EVALUATION

---

**Prompt for Personalization Dimension Weights Allocation**

You are an experienced evaluation expert for research articles. You excel at deeply understanding the goals, challenges, and key value points of a specific research task and the task initiator's persona, and then setting dynamic, reasonable, and well-justified weights for evaluation dimensions in subsequent personalized article assessments.
</system_role>

<user_prompt>
Here is a deep research task, as follows:
<task>
"{task_prompt}"
</task>

The user persona is as follows:
<persona>
"{persona_prompt}"
</persona>

<instruction>
Background: The research team will conduct in-depth and comprehensive research based on the above <task> and <persona> and ultimately produce a high-quality, personalized research article.
Your task: As the evaluation expert, you need to set the weights of the personalized evaluation criteria for this specific <task>. The evaluation will revolve around the following four dimensions:

1. **Goal Alignment:** Whether the research sufficiently and accurately understands the relationship between the task and the user persona, extracts deep and implicit needs, and generates a personalized report based on them.

2. **Content Alignment:** Whether the research selects and customizes content according to the user's interests, knowledge background, and preferences.

3. **Actionability & Practicality:** Whether the report is feasible, practical, and helpful for the user's decision-making.

4. **Presentation Fit:** Whether the report's language style, information structure, and presentation format match the user's cognitive habits and medium preferences.

Evaluation formula:
Total score = Goal Alignment * Goal Alignment weight + Content Alignment * Content Alignment weight + Actionability & Practicality * Actionability & Practicality weight + Presentation Fit * Presentation Fit weight. (Note: The sum of all weights must be exactly 1.0)

Core requirements:

1. **Deeply analyze the task and user persona:** Carefully study the specific content of <task>, its explicit goals, potential challenges, and hidden objectives. Combine this with <persona> to analyze the user's needs, background, and preferences, and understand the core value of the task's outcome.

2. **Dynamically assign weights:** Based on your analysis, assign weights to the four dimensions (use decimals between 0 and 1, e.g., 0.30). The key is to recognize that different tasks and personas emphasize different aspects, so weights must be flexibly adjusted according to task characteristics and persona, not fixed.

3. **Explain your reasoning:** Your analysis (<analysis>) must clearly and specifically explain why each dimension is assigned a given weight, and directly link your

---

reasoning to the requirements of <task> and the characteristics of <persona>. This is critical for evaluating the quality of your work.

4. **Output in the standard format:** Strictly follow the example format below: first output <analysis> with detailed reasoning, then immediately provide <json_output> with the weight assignment results.

</instruction>

<examples_rationale>
Below are two examples that demonstrate how to adjust the evaluation dimension weights and explain the reasoning based on changes in task nature and user persona. Focus on learning the thinking process and analytical method in these examples, not simply copying their content or weight values.
</examples_rationale>
...
Now strictly follow the above instructions and methodology, and start your work for the following task:
<task>
"{task_prompt}"
</task>

<persona>
"{persona_prompt}"
</persona>

Please output your <analysis> and <json_output>.

---

**Prompt for Goal Alignment Criteria Generation**

You are an experienced research article evaluation expert. You excel at breaking down abstract evaluation dimensions (such as "Goal Understanding and Personalization Insight") into actionable, clear evaluation criteria tailored to the specific research task and user persona, and assigning reasonable weights with explanations for each criterion.
</system_role>

<user_prompt>
**Background**: We are evaluating a research article written for the following research task under the dimension of Goal Alignment.
**Goal Alignment:** Whether the research fully and accurately understands the relationship between the task and the user persona, extracts deep and implicit needs, and generates a personalized report based on that understanding, with a focus on performing user-centered, deeply personalized matching between the user persona and task requirements.

<task>
"{task_prompt}"
</task>

The user persona is as follows:
<persona>
"{persona_prompt}"
</persona>

<instruction>
Your goal:
For the Goal Alignment dimension of this research article, formulate a set of detailed, specific, and highly targeted evaluation criteria that are tightly aligned with the above <task> and <persona>. You need to:

1. Deeply analyze the user persona and task scenario: Thoroughly examine the background characteristics, knowledge structure, cognitive habits, and latent expec-

tations of `<persona>`. Combine this with the specific application scenario of `<task>` to identify the user's core explicit needs and deeper implicit needs.

2. Formulate personalized evaluation criteria: Based on the above analysis, propose specific evaluation criteria that reflect a deep understanding of `<persona>` and a close fit to the `<task>` scenario. These criteria should assess whether the content is well adapted to the user persona in style, depth, perspective, and practicality.

3. Explain the personalization rationale: Provide a brief explanation (explanation) for each criterion, clarifying how it addresses the specific attributes of `<persona>` or special requirements of `<task>`, and why such targeting is critical to achieving a good match.

4. Assign rational weights: Assign a weight (weight) to each criterion, ensuring that the total sum is 1.0. The distribution of weights should directly reflect the relative importance of each criterion in measuring how well the content matches "this particular user" in "this particular task." The closer a criterion is tied to persona characteristics and task scenario, the higher its weight should be.

Core requirements:

1. Deep personalization orientation: The analysis, criteria, explanations, and weights must be deeply rooted in the uniqueness of `<persona>` (e.g., their professional background, cognitive level, decision-making preferences, emotional needs) and the specific context of `<task>`. Avoid generic or templated evaluation.

2. Focus on contextual responsiveness and resonance: The criteria should evaluate whether the content not only responds to the task at the informational level but also resonates with the context and expectations implied by the user persona in terms of expression style, reasoning logic, case selection, and level of detail.

3. Rationale must reflect targeting: The `<analysis>` section must clearly explain how key features were extracted from the given `<persona>` and `<task>` to form these personalized criteria. Each criterion's explanation must directly show how it serves this specific user and task.

4. Weights must reflect personalization priorities: The weight distribution must logically demonstrate which aspects of alignment are the most critical success factors for "this user" completing "this task."

5. Standard output format: Strictly follow the example format below. First output the `<analysis>` text, then immediately provide the `<json_output>`.

`</instruction>`

`<example_rational>`
The example below demonstrates **how to develop Goal Alignment evaluation criteria based on the task requirements**. Focus on understanding the **thinking process and analytical approach** used in the example, rather than simply copying its content or numerical weights.
`</example_rational>`
...

Please strictly follow the above instructions and methodology. Now, for the following specific task, start your work:
`<task>`
"{task_prompt}"
`</task>`

`<persona>`
"{persona_prompt}"
`</persona>`

Please output your `<analysis>` and `<json_output>`.
`</user_prompt>`

---

**Prompt for Content Alignment Criteria Generation**

You are an experienced research article evaluation expert. You are skilled at breaking down abstract evaluation dimensions (such as "Content Alignment") into actionable, clear, and specific evaluation criteria tailored to the given research task and user persona, and assigning reasonable weights and explanations for each criterion.
</system_role>
<user_prompt>
**Background**: We are providing a personalized scoring rubric for a specific task and user persona from the dimension of **Content Alignment**.
**Content Alignment**: Whether the research content is customized based on the user's interests, knowledge background, and other preferences.

<task>
"{task_prompt}"
</task>

The user persona is as follows:
<persona>
"{persona_prompt}"
</persona>
<instruction>
**Your Goal**: For the **Content Alignment** dimension of this research article, create a set of detailed, concrete, and highly tailored evaluation criteria for the above <task> and <persona>. You need to:

1. **Analyze the Task and Persona**: Deeply analyze <task> and <persona> to infer the user's potential interests, knowledge background, and the depth and breadth of content they may prefer.

2. **Formulate Criteria**: Based on your analysis, propose specific evaluation criteria that focus on whether the report's content matches the user's interest points and knowledge level.

3. **Provide Explanations**: For each criterion, provide a brief explanation (`explanation`) explaining why it is important for evaluating the content alignment for this <task>.

4. **Assign Weights**: Assign a reasonable weight to each criterion (`weight`), ensuring that the sum of all weights equals exactly 1.0. The weight allocation should logically reflect the personalization-first principle: criteria directly tied to unique personal traits, exclusive preferences, or specific contextual needs in the user persona should receive higher weights, as they are key to achieving true personalized content alignment.

5. **Avoid Overlap**: Make sure the evaluation criteria focus solely on the **Content Alignment** dimension, avoiding overlap with other dimensions such as Goal Alignment, Expression Style Alignment, and Practicality/Actionability.

**Core Requirements**:

1. **Strongly Linked to the Persona**: The analysis, criteria, explanations, and weights must be directly connected to the user's interests, knowledge background, or content preferences.

2. **Focus on Content Selection and Depth**: The criteria should assess whether the choice of content is precise and whether the depth is appropriate, rather than merely evaluating whether information is presented.

3. **Provide Sufficient Rationale**: The <analysis> section must clearly articulate the overall reasoning behind formulating these criteria and weights, linking them to <task> and <persona>. Each `explanation` must clarify why the individual criterion is relevant.

---

4. **Reasonable Weighting**: The weight distribution should be logical, reflecting the relative importance of each criterion in measuring content alignment, with particular emphasis on giving higher priority to personalized aspects.

5. **Standardized Output Format**: Strictly follow the format below — output the <analysis> text first, immediately followed by <json_output>.

</instruction>
<example_rational>
The following example demonstrates **how to formulate content alignment evaluation criteria based on the task requirements and user persona**. Pay close attention to the **thinking process and analytical approach** in this example, rather than simply copying the content or weight values.
</example_rational>
. . .
Please strictly follow the above instructions and methodology. Now, for the following specific task, start your work:
<task>
"{task_prompt}"
</task>
<persona>
"{persona_prompt}"
</persona>
Please output your <analysis> and <json_output>.
</user_prompt>

---

Scoring Prompt for Personalization

<system_role>You are a strict, meticulous, and objective expert in evaluating personalized research articles. You excel at deeply evaluating research articles based on specific personalization assessment criteria, providing precise scores and clear justifications.</system_role>

<user_prompt>
**Task Background**
You are given an in-depth research task. Your job is to evaluate a research article written for this task in terms of its performance in **"Personalization Alignment"**. We will evaluate it across the following four dimensions:

1. Goal Alignment

2. Content Alignment

3. Presentation Fit

4. Actionability & Practicality

<task>
"{task_prompt}"
</task>

**User Persona**
<persona>
"{persona_prompt}"
</persona>

**Article to be Evaluated**
<target_article>
"{article}"
</target_article>

**Evaluation Criteria**

You must evaluate the specific performance of this article in terms of personalization alignment, **following the criteria list below**, outputting your analysis and then assigning a score from 0–10. Each criterion includes its explanation, which you should read carefully.
<criteria_list>
{criteria_list}
</criteria_list>

<Instruction>
**Your Task**
Strictly follow **each criterion** in <criteria_list> to evaluate how <target_article> meets that criterion. You must:

1. **Analyze Each Criterion**: For each item in the list, think about how the article meets the requirements of that criterion.

2. **Analytical Evaluation**: Combine the article content, the task, and the user persona to analyze the article's performance for that criterion, pointing out both strengths and weaknesses.

3. **Scoring**: Based on your analysis, give a score between 0 and 10 (integer) for the article's performance on that criterion.

**Scoring Rules**
For each criterion, give a score between 0 and 10 (integer). The score should reflect the quality of the article's performance:

- 0–2 points: Very poor. Almost completely fails to meet the requirement.

- 2–4 points: Poor. Meets the requirement only partially, with significant shortcomings.

- 4–6 points: Average. Basically meets the requirement; neither particularly good nor bad.

- 6–8 points: Good. Mostly meets the requirement, with notable strengths.

- 8–10 points: Excellent/Outstanding. Fully or exceptionally meets the requirement.

**Output Format Requirements**
Strictly follow the <output_format> below to output the evaluation results for **each criterion**. **Do not include any irrelevant content, introductions, or conclusions**. Start from the first dimension and output all dimensions and their criteria in sequence:
</Instruction>

<output_format>

```
{
    "goal_alignment": [
        {
            "criterion": "[The text of the first Goal Alignment
                criterion]",
            "analysis": "[Analysis]",
            "target_score": "[integer score 0-10]"
        },
        {
            "criterion": "[The text of the second Goal Alignment
                criterion]",
            "analysis": "[Analysis]",
            "target_score": "[integer score 0-10]"
        },
        ...
    ],
    "content_alignment": [
        {
            "criterion": "[The text of the first Content Alignment
                criterion]",
```

```
            "analysis": "[Analysis]",
            "target_score": "[integer score 0-10]"
        },
        ...
    ],
    "presentation_fit": [
        {
            "criterion": "[The text of the first Presentation Fit
                criterion]",
            "analysis": "[Analysis]",
            "target_score": "[integer score 0-10]"
        },
        ...
    ],
    "actionability_practicality": [
        {
            "criterion": "[The text of the first Actionability &
                Practicality criterion]",
            "analysis": "[Analysis]",
            "target_score": "[integer score 0-10]"
        },
        ...
    ]
}
```

</output_format>

## J.2 PROMPTS IN QUALITY EVALUATION

**Prompt for Quality Dimension Weights Allocation**

You are an experienced expert in evaluating research reports. You excel at deeply understanding the goals, challenges, and core value points of a given research task, and setting dynamic, reasonable, and well-justified dimension weights for subsequent report quality evaluations.
</system_role>

<user_prompt>
Here is a deep research task as follows:
<task>
"{task_prompt}"
</task>

<instruction>
Background: The research team will conduct an in-depth and comprehensive investigation based on the above <task> and eventually produce a high-quality research report.
Your Task: As an evaluation expert, you need to set the evaluation dimension weights specifically for this <task>. The evaluation will be carried out around the following three dimensions:
1. Depth & Insight: Whether the report provides sufficient depth and unique insights.
2. Logical Coherence: Whether the report's reasoning framework is rigorous and its logical derivation coherent.
3. Clarity & Readability: Whether the report's language, information presentation, and formatting are clear and easy to understand, allowing readers to absorb the content smoothly.

Evaluation Formula: Total Score = (Depth & Insight * weight$_1$) + (Logical Coherence * weight$_2$) + (Clarity & Readability * weight$_3$). (Note: The sum of all weights must equal exactly 1.0)

Core Requirements:
1. Analyze the Task in Depth: Carefully study the <task> content, implicit objectives, potential challenges, and the core value of the deliverable.
2. Dynamically Allocate Weights: Based on your analysis, assign weights to the three dimensions (use decimals between 0 and 1, e.g., 0.4). The key is to understand that different tasks emphasize different aspects — weights must be adjusted flexibly based on task characteristics, rather than being fixed.
3. Explain the Allocation Rationale: Your analysis (<analysis>) must clearly and specifically explain why each dimension is given its corresponding weight and directly link your reasoning to the requirements and characteristics of <task>. This is the key criterion for evaluating your work quality.
4. Standardized Output Format: Strictly follow the example format below — first output the detailed rationale in <analysis>, then provide the weight allocation result in <json_output>.

</instruction>

<examples_rationale>
Below are two examples, which demonstrate how to adjust dimension weights according to the nature of the task and explain the reasoning. Please focus on learning the thinking process and analytical approach shown in the examples, rather than simply copying their content or numerical values.
</examples_rationale>
. . .
Please strictly follow the above instructions and methodology. Now, for the following specific task, begin your work:
<task>
"{task_prompt}"
</task>
Please output your <analysis> and <json_output>.
</user_prompt>

---

## Prompt for Depth & Insight Criteria Generation

You are an experienced expert in evaluating research reports. You excel at breaking down abstract evaluation dimensions (such as "Depth & Insight") into actionable, task-specific, and clear criteria, assigning reasonable weights and explanations for each.
</system_role>

<user_prompt>
**Background**: We are evaluating a research report based on three dimensions: Depth & Insight, Logical Coherence, and Clarity & Readability.

1. **Depth & Insight:** Whether the report provides sufficient depth and unique insights.

2. **Logical Coherence:** Whether the report's reasoning framework is rigorous and its logical derivation coherent.

3. **Clarity & Readability:** Whether the report's language, information presentation, and formatting are clear and easy to understand.

<task>
"{task_prompt}"
</task>

<instruction>
**Your Goal**: For the **Depth & Insight** dimension of this report, develop a detailed, specific, and highly task-targeted set of evaluation criteria. You need to:

1. **Analyze the Task**: Examine `<task>` in depth and identify where deep analysis, logical reasoning, insight extraction, or value judgment are required to demonstrate insight.

2. **Formulate Criteria**: Based on the analysis, propose concrete evaluation criteria focusing on analytical depth, logical rigor, originality, and value of conclusions.

3. **Explain Each Criterion**: Provide a brief explanation (`explanation`) for why this criterion is important for evaluating Depth & Insight for `<task>`.

4. **Assign Weights**: Assign a reasonable weight (`weight`) to each criterion, ensuring the weights sum exactly to **1.0**. The weights should reflect the relative importance of each criterion within the Depth & Insight dimension.

5. **Avoid Overlap**: Clearly focus only on criteria relevant to **Depth & Insight**, avoiding aspects of **Logical Coherence** (structure) or **Clarity & Readability** (language, formatting).

**Core Requirements**:

1. **Stay Task-Specific**: The analysis, criteria, explanations, and weights must directly relate to the task's core requirements and characteristics.

2. **Go Beyond the Surface**: The criteria should assess analytical depth, reasoning rigor, originality of insights, and value of conclusions — not just listing information.

3. **Provide Strong Rationale**: The `<analysis>` section must clearly explain the overall approach to designing the criteria and weights, linking it to `<task>`. Each `explanation` must justify the criterion.

4. **Ensure Reasonable Weighting**: Weight distribution must be logical, reflecting the relative importance of each criterion in showing insight.

5. **Standardized Output Format**: Strictly follow the format below: output `<analysis>` first, then `<json_output>`.
</instruction>

<example_rational>
Below is an example demonstrating **how to design Depth & Insight criteria**. Focus on the **thinking logic and analytical approach** rather than copying its contents or weight numbers.
</example_rational>
. . .

Please strictly follow the above instructions and methodology. Now, for the following specific task, begin your work:
<task>
"{task_prompt}"
</task>
Please output your `<analysis>` and `<json_output>`.

---

### Scoring Prompt for Quality

```
<system_role>
```
You are a strict, meticulous, and objective expert in evaluating the quality of research articles. You excel at deeply evaluating research articles based on specific quality assessment criteria, providing precise scores and clear justifications.
```
</system_role>

<user_prompt>
```

**Task Background**

You are given an in-depth research task. Your job is to evaluate a research article written for this task. We will evaluate it across the following three dimensions: Depth & Insight, Logical Coherence and Clarity & Readability. The task is as follows:

```
<task>
"{task_prompt}"
</task>
```

**Article to be Evaluated**

```
<target_article>
"{article}"
</target_article>
```

**Evaluation Criteria**

You must evaluate the article's performance for each criterion in the list below, outputting your analysis and then assigning a score from 0–10. Each criterion includes its explanation, which you should read carefully.

```
<criteria_list>
{criteria_list}
</criteria_list>

<Instruction>
```

**Your Task**

Strictly follow each criterion in `<criteria_list>` to evaluate how `<target_article>` meets that criterion. You must:

1. **Analyze Each Criterion**: For each item in the list, think about how the article meets the requirements of that criterion.

2. **Analytical Evaluation**: Combine the article content with the explanation of the criterion to analyze the article's performance for that criterion, pointing out both strengths and weaknesses.

3. **Scoring**: Based on your analysis, give a score between 0 and 10 (integer) for the article's performance on that criterion.

**Scoring Rules**

For each criterion, give a score between 0 and 10 (integer). The score should reflect the quality of the article's performance:

- 0–2 points: Very poor. Almost completely fails to meet the requirement.

- 2–4 points: Poor. Meets the requirement only partially, with significant shortcomings.

- 4–6 points: Average. Basically meets the requirement; neither particularly good nor bad.

- 6–8 points: Good. Mostly meets the requirement, with notable strengths.

- 8–10 points: Excellent/Outstanding. Fully or exceptionally meets the requirement.

**Output Format Requirements**

Strictly follow the `<output_format>` below to output the evaluation results for **each criterion**. **Do not include any irrelevant content, introductions, or conclusions**. Start from "criterion 1" and output all criteria in order:

```
</Instruction>

<output_format>

{
    "depth_insight": [
        {
            "criterion": "[The text of the first Depth & Insight
                criterion]",
```

```
            "analysis": "[Analysis]",
            "target_score": "[integer score 0-10]"
        },
        {
            "criterion": "[The text of the second Depth & Insight
                criterion]",
            "analysis": "[Analysis]",
            "target_score": "[integer score 0-10]"
        },
        ...
    ],
    "logical_coherence": [
        {
            "criterion": "[The text of the first Logical Coherence
                criterion]",
            "analysis": "[Analysis]",
            "target_score": "[integer score 0-10]"
        },
        {
            "criterion": "[The text of the second Logical
                Coherence criterion]",
            "analysis": "[Analysis]",
            "target_score": "[integer score 0-10]"
        },
        ...
    ],
    "clarity_readability": [
        {
            "criterion": "[The text of the first Clarity &
                Readability criterion]",
            "analysis": "[Analysis]",
            "target_score": "[integer score 0-10]"
        },
        {
            "criterion": "[The text of the second Clarity &
                Readability criterion]",
            "analysis": "[Analysis]",
            "target_score": "[integer score 0-10]"
        },
        ...
    ]
}

</output_format>
```

## J.3 PROMPTS IN RELIABILITY EVALUATION

**Prompt for Claim Extraction**

You will see a research report, and your task is to extract only all verifiable factual statements (factual claims) from the text.
# Definition of Factual Statement A factual statement is a verifiable claim about the objective state of the external world. It describes facts that have already occurred, quantifiable data, recognized classifications, or scientific laws — not the author's subjective opinions, intentions, plans, or predictions about the future, nor descriptions about the report's own plans or structure.
# Guidelines for Identifying Factual Statements
# Types to Extract (Examples)

- Specific data and statistics: "In 2023, global electric vehicle sales reached 14.1 million units."

- Past historical events: "The company was founded in Shanghai, China, in 2010."

- Recognized classifications or definitions: "Li Qiang constructed a socioeconomic status index (SES) based on income, education, and occupation, dividing society into seven classes [15]."

- Cited research findings: "Studies show that more than eight hours of sleep are critical for memory consolidation [8]."

# Types to Exclude (Examples)

- Goals and intentions: Any statement describing the "purpose," "goal," or "aim" of this document or project.
    - Example: "The goal of this report is to systematize personal creative activities." or "This project aims to verify the small-revenue model."

- Plans and proposals: Plans about future actions, strategies, or content.
    - Example: "The content pillars include: travel sketch diaries, process breakdowns, tool reviews..." or "We will execute this plan in three phases."

- Self-referential statements about the document: Statements introducing the report's structure or content.
    - Example: "This report is a three-month brand and operations execution manual for..." or "Chapter 3 will discuss market analysis in detail."

- Predictions and speculations: Estimations or guesses about what might happen in the future.
    - Example: "This strategy is expected to increase user stickiness by 20%." or "This could create new business opportunities."

- Opinions and recommendations: The author's subjective judgments, opinions, or suggestions.
    - Example: "We believe this is a key breakthrough." or "Therefore, we recommend adopting Plan A."

- Research methods: Descriptions of how the research or work will be conducted.
    - Example: "This study will adopt a mixed-method approach combining qualitative and quantitative analysis."

# Extraction Rules and Output Format For each factual statement you find, determine whether it includes a reference citation, and extract it as a (fact, ref_idx, url) triple. Citations in the text may appear in the following forms:

1. A piece of text + space + number, for example: "Li Qiang constructed a socioeconomic status index (SES) based on income, education, and occupation, dividing society into seven classes 15"

2. A piece of text + [number(s)], for example: "Li Qiang constructed a socioeconomic status index (SES) based on income, education, and occupation, dividing society into seven classes [15]"

3. A piece of text + [number(s)†(some line numbers etc.)], for example: "Li Qiang constructed a socioeconomic status index (SES) based on income, education, and occupation, dividing society into seven classes [15†L10][5L23][7†summary][9summary]"

4. [Cited source](citation link), for example: "According to [ChinaFile: A Guide to Social Class in Modern China](https://www.chinafile.com/reporting-opinion/media/guide-social-class-modern-china), Chinese society can be divided into nine classes"

When extracting, pay attention to the following:

1. The extracted fact should be a complete, understandable statement — not just a phrase or fragment.

2. If a fact cites multiple references, output multiple triples. For example, if it cites two references, output (fact, ref_idx_1, url_1) and (fact, ref_idx_2, url_2).

3. For the third form of citation, only take the first numeric part as ref_idx, ignoring indicators of specific locations. For the fourth form (where the source and link appear directly in the text), set ref_idx uniformly to 0.

4. If a factual statement has no citation, set both ref_idx and url to empty strings "".

Output Requirements: You should return a JSON list, where each item is one triple. For content you are unsure about, err on the side of caution — it's better to miss something than to mislabel it. If there are no factual statements in the article, return an empty list [].
JSON Example:

```
[
    {
        "fact": "Text from the original article, use full-width
            Chinese quotation marks, escape English quotes with a
            single backslash",
        "ref_idx": "The index of the cited reference in the
            reference list for this statement; leave empty if none",
        "url": "The link of the cited reference (extracted from
            the report's reference list or from the inline
            citation), leave empty if none"
    },
    {
        "fact": "In 2023, global electric vehicle sales reached
            14.1 million units.",
        "ref_idx": 12,
        "url": "https://iea.org/reports/global-ev-outlook-2024"
    },
    {
        "fact": "Tesla went public on NASDAQ in 2010.",
        "ref_idx": "",
        "url": ""
    },
    {
        "fact": "Studies show that more than eight hours of sleep
            significantly enhances memory consolidation.",
        "ref_idx": 5,
        "url": "https://doi.org/10.1016/j.neurobiol.2020.101945"
    },
    {
        "fact": "According to UNEP
            (https://www.unep.org/resources/emissions-gap-report-2023),
            global greenhouse gas emissions reached a record high
            in 2023.",
        "ref_idx": 0,
        "url":
            "https://www.unep.org/resources/emissions-gap-report-2023"
    }
]
```

Below is the main text of the research report: {report_text}
Now start extracting, and directly output the JSON list — do not output any small talk or explanation.

