# OpenReview forum: "Towards Personalized Deep Research: Benchmarks and Evaluations"
_ICLR.cc/2026/Conference — ICLR 2026 Poster_

### Official Review · Reviewer_vdTb · 2025-10-30

**Soundness:** 2
**Presentation:** 2
**Contribution:** 2
**Rating:** 2
**Confidence:** 3

**Summary:**

The paper proposes a new benchmark for personalized deep research. Deep research is a popular mode of AI agents, where an agent utilizes several tools like iterative search, to answer complex question. Personalization in the context of deep research is underexplored. This paper presents a new benchmark to evaluate the personalization capabilities of LLMs in the context of deep research. Specifically, the paper focuses on evaluating 3 key aspects of deep research responses -- personalization, response quality, and reliability. The paper presents automated metrics to evaluate each of these dimensions.

**Strengths:**

1. The paper focuses on a relatively underexplored aspect of LLM evaluation -- personalization in deep research. The paper discusses the key drawbacks of existing benchmarks and builds on those aspects.
2. The paper collects personalization data in a thorough manner by including people of diverse backgrounds and collect data from these folks over prolonged interactions.

**Weaknesses:**

1. I'm concerned about the privacy implications of the benchmark release. According to Table 5 in Appendix E, the persona collection process includes identity characteristics like name, age, gender, etc. These information along with others could often be enough to reveal the identity of an individual.

2. The proposed evaluation process seems to be weak. Along the personalization dimension in Section 4.1, the authors ask the LLM to generate both the criteria and the weight of each criteria. Were these information reviewed by experts? Also, it seems unclear which LLM was used for this specific task.

3. The paper misses out on key details in some sections like 4.2. Why did the paper choose not to use well-established factuality metrics like FActScore [1]? How are the verifiable claims extracted from the answers?

4. How is $\lambda_a$ in Equation 6 chosen?

5. Most of the results presented in the paper are scores generated by LLM. In Section 5.5, the paper evaluates the correctness of these scores by comparing it human annotators. However, this task was simplified to a great degree where the annotator had to choose between a couple of responses. The paper compares LLM score agreement in this setting. This is not an accurate measure of correlation of the actual score (which are between 1-10). Even in this simplified setting, the correlation seems to be poor raising questions about the quality of LLM ratings.

6. In Table 1, 3, and left figure of Figure 3, we observe that incorporating additional personal context doesn't change the performance too much, even the P-Score. Does this mean current LLMs aren't able to personalize based on the provided context? If not, how can the P-score be so high without any context?


[1] FActScore: Fine-grained Atomic Evaluation of Factual Precision in Long Form Text Generation

**Questions:**

Please respond to the questions in the above sections.

**Details Of Ethics Concerns:**

I have concerns that releasing this dataset might compromise the privacy of the study participants.

---

> ### Author Response · Authors · 2025-11-19
> **Response to Reviewer vdTb (Q1, Q2)**
>
> Dear reviewer vdTb,
>
> Many thanks for your detailed and constructive comments.
>
> ---
>
> **Q1: Concerned about the privacy implications**
>
> We thank the reviewer for raising this critical point and fully agree that privacy is essential. We want to clarify that all user profiles were collected under informed consent and underwent strict anonymization prior to release (Sec. 7), and the publicly released benchmark contains only fully de-identified and abstracted derivatives of these profiles, not the original user data. The structured persona schema (Appendix D) was designed for research documentation only, and the released benchmark contains no identifiable information — names were replaced with pseudonyms, ages were generalized to ranges (e.g., “25–30”), and any sensitive attributes (e.g., occupation, education, hometown) and overly detailed descriptions were abstracted, obfuscated or generalized (e.g., “a university in Jiangsu”, “a famous internet company”) to prevent any possibility of re-identification.
>
> One example in our dataset: { "Name": "Li Chen" (pseudonym), "Age": "19–21", "Gender": "Male", "Occupation": "Undergraduate student majoring in computer science", "Location": "Eastern China region", "Education": "Studying at a university in Jiangsu", ...}.
>
> We further confirm that the dataset contains absolutely no personally identifiable information (PII) that could pinpoint an individual such as real names, phone numbers, email addresses, ID numbers, or detailed residential addresses.
>
> ---
>
> **Q2: Concerned about llm evaluation process**
>
> We acknowledge that the automatic generation of evaluation criteria and weights has inherent limitations. To enhance reliability, we conducted multiple rounds of prompt refinement: in each round, we generated criteria and weights, manually reviewing them (ensure the criteria were reasonable and interpretable, sufficiently discriminative across different personas and tasks, and the criteria focused strictly on their respective dimensions without semantic overlap). Based on the review feedback, the prompt was optimized and re-tested. We conducted 5 rounds of iteration, a stable prompt was finalized for batch generation (Section I).
>
> Specially, in code implement, we adopted repeated sampling and averaging for weight generation instead of relying on a single LLM output: each set of weights was generated 5 times and aggregated (averaging + normalization + adjustment). This reduces randomness, improves reliability, and avoids outlier influence. The final results therefore reflect stable consensus values rather than single stochastic outputs.
>
> In addition, we experimentally compared several LLMs for generation—GPT-4o, GPT-5, Gemini-2.5-Pro, and Claude-3.7—and found GPT-5 produced the most consistent and high-quality results. Finally, GPT-5 was therefore selected for generating criteria and weights in Sections 4.1 and 4.2. Finally, we emphasize that the generated criteria serve as structured, persona-grounded rubrics rather than subjective or model-specific preferences. The purpose of automation here is to standardize evaluation across diverse personas and tasks, while expert validation of the prompt ensures that the generated rubrics remain coherent, interpretable, and aligned with the intended evaluation constructs.

---

> ### Author Response · Authors · 2025-11-19
> **Response to Reviewer vdTb (Q3)**
>
> Dear reviewer vdTb,
>
> Many thanks for your detailed and constructive comments.
>
> ---
>
> **Q3：Factuality evaluation design**
>
> Thank you for your thoughtful review and questions regarding our factuality evaluation.
>
> 1. Why we chose not to use well-established factuality metrics like FActScore
>
> We developed our Factual Reliability (R) metric instead of using FActScore due to fundamental differences in evaluation goals:
> - FActScore assesses factual accuracy against static, external knowledge bases (like Wikipedia). In contrast, DRAs must produce evidence-based reports. Our metric, Factual Reliability (R), is specifically designed to measure citation compliance—ensuring every claim in the report is supported by the specific sources cited within the report itself.
> - Research reports often contain specialized or emerging facts not covered by static knowledge sources. Our method, by checking the report's self-cited references, better aligns with the professional standard required for verifying claims in a research context.
>
> We also provide a comparison between our our method and FActScore:
> | Feature | Our Method (Factual Reliability R) | FActScore |
> |---------|------------------------------------|-----------|
> | Primary Evaluation Goal | Factual reliability and citation compliance of deep research reports | General factual precision in long-form text generation (e.g., biographies) |
> | Granularity | Verifiable Claim | Atomic Fact |
> | Verification Source | The report's self-cited references (links or indices provided by the report) | Independent external knowledge sources (e.g., Wikipedia, pre-defined KBs) |
> | Extraction Method | Instruction-tuned LLM as a fact extractor, outputting a triple containing fact, index, and URL | Decomposes generated text into atomic facts, verified using retrieval and classification models |
> | Core Focus | Whether the claim is cited and supported by the cited source | Whether the claim is supported by the general knowledge source |
>
> 2. How the verifiable claims are extracted from the answers (Section 4.2)
>
> The process for extracting Verifiable Claims used for assessment in Section 4.2 employs an LLM (GPT-5-Mini, Prompt in Appendix I.3) as a extractor. The specific steps are:
> - Role Instruction: The LLM is first instructed to act as a "claim extractor".
> - Extraction Task: The LLM is tasked with analyzing the full report text generated by the agent and extracting all verifiable claims (i.e., factual statements).
> - Output Format: The extractor formats the results into a JSON list, where each element is a triple containing three specific fields:
>     - "fact": The exact text from the original article itself, representing the verifiable claim.
>     - "ref_idx": The index of the cited reference in the report's reference list (left empty if no inline citation is present).
>     - "url": The link of the cited reference (left empty if no link is available).
>
> This ensures each claim retains its original citation context for subsequent verification. (i.e., verifying if the claim is cited and supported by its cited source).
>
> We appreciate the reviewer for pointing this out. We will include a discussion of FActScore in the paper and clarify its distinction from our citation-grounded verification.

---

> ### Author Response · Authors · 2025-11-19
> **Response to Reviewer vdTb (Q4, Q5, Q6)**
>
> Dear reviewer vdTb,
>
> Many thanks for your detailed and constructive comments.
>
> ---
>
> **Q4：How is λ in Equation 6 chosen?**
>
> We apologize for this confuse. In Equation (6), the aggregation weights λₐ (for P, Q, and R) were uniformly set equal to 1 in all reported experiments, reflecting our assumption that personalization, quality, and reliability are equally important. In practice, this is equivalent to taking the average of the three scores, since any constant scaling of λₐ yields the same normalized result as long as all weights are equal. We will revise the equation (6) in the manuscript to make this formulation clearer.
>
> ---
>
> **Q5：Reliability of LLM-generated scores**
>
> We appreciate the reviewer’s concern regarding the reliability of LLM-generated scores. We would like to clarify that our validation in Section 5.5 evaluates both directional agreement and score-level consistency, rather than relying solely on a simplified pairwise setting.
>
> First, Pairwise Comparison Agreement (PCA) assesses whether the LLM judge and human experts agree on which report is better for each criterion. This metric evaluates the discriminative reliability of the judge—i.e., whether it captures the relative quality differences that matter for ranking models.
>
> Second, Mean Absolute Rating Deviation (MARD) directly measures alignment on the 1–10 continuous scores used in our benchmark. Unlike correlation coefficients, which are unstable under sparse or noisy human ratings, MARD provides a robust estimate of numeric score deviation, and is widely used in evaluating long-form judgment alignment.
>
> Although the highest PCA observed is 0.43, this value is reasonable given the nature of the task. Persona-conditioned deep research involves multiple interacting dimensions, including personalization, reasoning quality, factuality, and citation alignment. Because annotators must jointly evaluate all of these aspects when scoring long-form outputs, even trained human experts can reasonably disagree on fine-grained numeric ratings in such multi-criteria judgments. Therefore, perfect alignment is neither expected nor observed even among human judges. The PCA values should be interpreted as relative agreement across judge models, not as an absolute indicator of task simplicity.
>
> Based on the combined factors, we selected GPT-5 as the judge LLM that best balances directional consistency and score stability.
>
> ---
>
> **Q6：Why Does P-Score Remain High Even Without Persona Context?**
>
> We thank the reviewer for this insightful observation, please allows us to clarify a key design of our benchmark and the corresponding results. A user profile in our work comprises two distinct parts: explicit personas (structured, static attributes like demographics and preferences) and personalized context (unstructured, dynamic interaction data, contain much more irrelevant information). The experiments are designed to test these components systematically: Table 1 presents the main results under the Task w/Persona setting, where agents receive explicit personas. Table 2 and Figure 3 compare three configurations—Task Only (no profile), Task w/Context (only dynamic context), and Task w/Persona (explicit persona)—while Table 3 specifically explores enhancing the Task w/Context setting with memory systems.
>
> Your question is a valuable observation. Yes, current LLMs still show limited ability to perform true personalization based solely on unstructured context. As shown in Table 2 and Figure 3, Task w/ Context yields only marginal improvements over Task Only and less gains compared to Task w/Persona, indicating that models often fail to extract or leverage relevant user-specific cues from noisy, unstructured histories. The relatively high P-scores in Table 1 are achieved precisely because agents were provided with explicit personas, which offer a clear and strong signal for adaptation.
>
> Finally, concerning the baseline P-Score under Task Only likely stem from the model achieving basic task relevance by making plausible, generic assumptions inferred from the task description itself—for example, when asked “analyze recent AI policy trends,” the model, without any user information, may assume the user is a general researcher or policymaker and produce a broad overview covering global regulations and ethical discussions. Such a response could coincidentally align with parts of the user’s interests, leading to moderate relevance. However, when provided with persona information (e.g., a policy analyst focusing on European AI governance), the model can tailor its response toward region-specific policies, key EU directives, and professional insights, demonstrating true, profile-driven personalization. The significant score improvement with explicit personas (Table 2) thus confirms that the P-score is sensitive to and effectively captures the added value of genuine personalized adaptation.

---

> ### Author Response · Authors · 2025-11-27
>
> Dear Reviewer vdTb,
>
> We highly appreciate the constructive comments and insightful suggestions you have offered for our work. Your comments have been extremely valuable in helping us improve the clarity and quality of the paper.
>
> As the discussion period has only a few days left, in order for us to have sufficient time to address any additional questions you may have, we kindly encourage you to engage in the ongoing discussion and share any further insights or clarifications you may have.
>
> If our responses have addressed your concerns, we would be sincerely grateful if you could consider updating your score accordingly.
>
> Thank you very much for your time and consideration. We truly appreciate your support and look forward to hearing from you.

---

### Official Review · Reviewer_8bjw · 2025-10-30

**Soundness:** 2
**Presentation:** 2
**Contribution:** 2
**Rating:** 4
**Confidence:** 3

**Summary:**

The paper introduces a benchmark for personalized deep research agents with 25 real human participants created persona profiles. The authors designed a new evaluation framework for this benchmark that jointly measures personalization, content quality and factuality.

**Strengths:**

1. Real human persona and questions and tasks are designed by human expert that increases the credibility of the benchmark in reflecting real user behaviors.

2. Comprehensive analysis across different models and agents.

**Weaknesses:**

1. The motivation for combining personalization and deep research into one benchmark is lacking. These two aspects are largely orthogonal to me as they measure different things. The authors did not explain why a fused benchmark is necessary. One can imagine that combining the scores of existing personalization and deep-research benchmarks are already a good metric for measuring personalized deep research agent. There is no correlation analysis showing that performance on this benchmark differs from performance on either personalization benchmarks. For example, I want to see that a model performs well on both personalization and deep research agent benchmark, but performs poorly on your benchmark. Without such evidence, the benchmark’s purpose and contribution remain weak.

2. LLM as judge bias. For personalization benchmark, it's important to show that llm as judge are unbiased, potentially by comparing with the real participants' and users' feedback on some models, so you can show that llm evaluation are aligned with actual user behaviors. The section 5.5 has a alignment with some human rating, this does not address the problem, since it's a personalization benchmark rather than a rlhf benchmark. It would be great to see the actual alignment with the actual user who generated the persona.

3. Scale is somewhat limited. With only 25 profiles, it's hard to generalize to the whole populations and demographics.

**Questions:**

1. Can you show the correlation of your benchmark with some personalization benchmark and research agent benchmark?

2. How to ensure that llm as judge are aligned with users behaviors?

---

> ### Author Response · Authors · 2025-11-19
> **Response to Reviewer 8bjw (W1 & Q1)**
>
> Dear reviewer 8bjw,
>
> Many thanks for your detailed and constructive comments.
>
> ---
>
> **W1 & Q1: Motivation, Necessity For This Bench, and Comparison with Existing Benchmarks**
>
> We thank the reviewer for this fundamental question.
>
> 1. Synergistic Necessity
>
> We contend that in a genuine PDR scenario, the two dimensions are inseparably integrated. The agent must utilize the specific user persona not merely as a boundary condition, but as a dynamic filter and driver throughout the complex, multi-step deep research process.
> - Example: In a "Study Abroad" task, a generic research agent might offer a static list of top-ranked schools. In contrast, a PDR agent conditions the entire planning–retrieval–analysis pipeline on the user—prioritizing scholarships and ROI for a budget-conscious student versus industry networks for a career-transitioning professional.
>
> 2. Quantitative Evidence: Low Correlation with Traditional Benchmarks
>
> To quantitatively address the concern that a simple combination of existing benchmark scores suffices, we compared the performance rankings of three common models on our PDR Bench against scores from a representative Traditional Personalization Benchmark (PersonaFeedback) and a Traditional Deep Research Benchmark (DeepResearch Bench).
>
> | Model             | PDR (PQR) | PersonaFeedback (P) | DeepResearch (DR) | P + DR     |
> | ----------------- | ------------- | ------------------- | ----------------- | ---------- |
> | Gemini-2.5-Pro    | 5.53      | 79.6                | 35.12             | 114.72     |
> | Claude-3.7-Sonnet | 4.83          | 77.8                | 40.67         | 118.47 |
> | GPT-4.1           | 4.28          | 80.0            | 33.46             | 113.46     |
>
> Key Findings:
> - The rankings on PDR do not align with traditional benchmarks. GPT-4.1 ranks 1st in personalization but last in PDR. Claude-3.7 ranks 1st in deep research but 2nd in PDR.
> - The model with the highest simple sum score (Claude, 118.47) is not the top PDR performer (Gemini).
> - Weak Correlations: Spearman rank correlations are low ( ρ(PDR, PersonaFeedback) ≈ -0.5, ρ(PDR, DeepResearch Bench) ≈ 0.5, ρ(PDR, P + DR) ≈ 0.5).
>
> These weak correlations confirm that PDR performance cannot be approximated by combining existing benchmark scores. Instead, PDR evaluates a distinct capability: persona-conditioned long-horizon deep research.
>
> 3. Landscape Analysis: Comparison with Related Benchmarks
>
> We appreciate the reviewer’s request. Below is a broader comparison with representative personalization and deep-research benchmarks:
>
> | Benchmark            | Category                       | Persona                    | Personalization Eval     | Long-form Research                        | Scale                   | Evaluation    |
> |----------------------|---------------------------------|----------------------------|---------------------------|--------------------------------------------|---------------------------|----------------|
> | PersonaFeedback      | Personalization                | ✓ Explicit persona         | ✓ Human-alignment        | ✗ Short responses, no multi-step research | ~8,298 samples          | Human gold     |
> | PersonaLens          | Personalization                | ✓ Multi-session persona    | ✓ Task + personalization | ✗ No long-form research                   | 111 tasks/20 domains    | LLM + human    |
> | PersonaGym           | Personalization                | ✓ Behavioral persona       | ✓ PersonaScore           | ✗ Dialogue not research                   | 10,000 Qs               | LLM + human    |
> | LaMP                 | Personalization                | ✓ History-based adaptation | ✓ Multi-task eval        | ✗ Short-form tasks                        | 7 task types            | Hybrid         |
> | DeepResearch Bench   | Deep Research                  | ✗                          | ✗                        | ✓ Cross-domain deep research              | 100 queries             | LLM-judge      |
> | ResearcherBench      | Deep Research                  | ✗                          | ✗                        | ✓ Scientific deep research                | 65 AI queries           | Experts rubic + LLM  |
> | DeepResearchGym      | Deep Research                  | ✗                          | ✗                        | ✓ Reproducible research                   | Sandbox                 | LLM-judge      |
> | PDR-Bench (Ours) | Personalized Deep Research | ✓ Explicit & dynamic   | ✓ P-Score            | ✓ Full deep-research pipeline         | 250 queries (25×50) | LLM-judge  |
>
>
> As shown, personalization benchmarks emphasize preference alignment in short contexts, while research benchmarks focus on depth without user information. PDR-Bench is the first to bridge this gap, explicitly evaluating whether agents can synthesize information and adapt reasoning strategies based on user profiles to deliver customized long-form deep research reports.

---

> > ### Comment · Reviewer_8bjw · 2025-11-26
> >
> > Thank you for addressing my questions, I am raising my score accordingly.

---

> > > ### Author Response · Authors · 2025-11-27
> > > **Thanks for your feedback**
> > >
> > > Dear Reviewer 8bjw,
> > >
> > > Thank you very much for your positive feedback and for taking the time to reconsider our work. We sincerely appreciate your thoughtful review and constructive suggestions, which have helped us improve the clarity and quality of the paper. We are grateful for your recognition and the score adjustment.
> > >
> > > Thank you again for your time and effort.

---

> ### Author Response · Authors · 2025-11-19
> **Response to Reviewer 8bjw (W2, W3, Q2)**
>
> Dear reviewer 8bjw,
>
> Many thanks for your detailed and constructive comments.
>
> ---
>
> **W2: Bias of LLM as judge**
>
> We appreciate the reviewer’s insightful concern regarding potential bias in LLM-as-judge evaluation for personalization tasks. Our benchmark evaluates persona-conditioned reasoning correctness rather than subjective personal preference. The goal is to assess whether a model can accurately follow a persona’s goals, constraints, and decision criteria throughout a multi-step deep-research process. We therefore rely on trained expert annotators, who are better equipped to evaluate multi-step reasoning quality, evidence grounding, and adherence to persona-defined objectives in a consistent and rubric-guided manner.
>
> To ensure stability and reduce stochastic variance in LLM-based evaluation, we adopt a three-run evaluation protocol, where each judgment is independently generated three times and averaged. This approach is widely used in recent LLM evaluation work to mitigate sampling noise and improve score reliability. Combined with our expert-annotated human evaluations (Sec. 5.5), this provides a consistent and reproducible assessment pipeline for PDR tasks.
>
> We agree that involving persona-matched real users can be valuable as a supplementary perspective, especially for understanding how personalized reasoning aligns with lived user experience. While such alignment is not strictly required for rubric-based persona-grounded correctness evaluation, we plan to explore a small-scale user study in future extensions to complement our current expert and LLM-based evaluation.
>
> ---
>
> **W3: Dataset Scale**
>
> We appreciate the reviewer’s suggestion. For this initial release, our priority was the authenticity and depth of each user profile, including both explicit attributes and dynamic interaction contexts, which required intensive human annotation (5,939 instances; 85 person-days of work and a budget of approximately $6,000 USD), quality control is essential at this pilot stage.
>
> Although there are 25 personas, pairing each deep-research task with 5 distinct profiles results in 250 personalized task-persona combinations. This design ensures every task is evaluated across multiple heterogeneous user needs, substantially enriching personalization variability.
>
> Furthermore, these 25 profiles were designed for maximum diversity across key real-world parameters, including:
> - Age range: Teenagers to retirees (70s).
> - Profession diversity: Spanning engineering, finance, humanities, education, medicine, etc.
> - Socioeconomic variability: Low-income students to high-income executives.
> - Life goals: Academic advancement, career transitions, investment, health, parenting, etc., plus balanced gender representation.
>
> It is important to note that due to the high computational costs of Deep Research Agents (DRAs), most existing benchmarks feature <= 100 queries. Our dataset of 250 unique personalized queries (25 personas x 50 tasks) represents a leading scale for this complex evaluation paradigm. This scale provides ample scope for rigorous statistical analysis.
>
> We view this as a pilot-scale benchmark establishing the methodological foundation, with plans to expand the dataset to multiple languages and cultural backgrounds in future work.
>
> ---
>
> **Q2：How to ensure that llm as judge are aligned with users behaviors?**
>
> Thank you for the insightful question. PDR does not evaluate subjective personal preference or behavioral prediction. Instead, it measures persona-conditioned reasoning correctness—whether the model adheres to persona-specified goals, constraints, and decision criteria within a deep-research workflow. This is an objective rubric-guided evaluation, not modeling real user behavior variance.
>
> Personas originate from real users, but real users may lack expertise to evaluate complex research report. Thus, trained expert annotators define the rubric and LLM-as-judge approximates expert decisions. As shown in Sec. 5.5, LLM-as-judge correlates well with expert scores in the personalized setting, supporting reliability and reproducibility.
>
> To enhance stability, each judgment is sampled three times and averaged, reducing randomness and improving consistency.
>
> We agree that studying alignment with actual behavioral outcomes is meaningful and plan future small-scale persona-matched studies. However, for correctness-oriented evaluation, expert-aligned rubric evaluation provides a practical and methodologically sound foundation.

---

### Official Review · Reviewer_maAF · 2025-10-31

**Soundness:** 3
**Presentation:** 3
**Contribution:** 3
**Rating:** 6
**Confidence:** 3

**Summary:**

Introduces a new benchmark and evaluation framework to address a gap in assessing Personalization in Deep Research Agents.
Personalized Deep Research Bench is a benchmark specifically designed to evaluate personalization in DRAs. Contains 250 user research task pairs.  PQR Evaluation Framework focuses on assessing the personalization alignment of a DRA output given a persona. Incorporates Goal Alignment, Content Alignment, Presentation Fit, and Actionability & Practicality.

**Strengths:**

The idea here is good. Objective measures of Research Agent outputs are obviously important, but there is value in customizing the research process and the resulting artifact to match the sensibilities of the user. So the work addresses this particular gap.

I expect the benchmark itself would be of some value to the research community. In particular the inclusion of structured and context data in user profiles, and the diversity of tasks.

The personalization metric is well defined and the evaluations presented are comprehensive.

**Weaknesses:**

It would be better if the personas were developed from real world interactions, the authenticity is limited. The sample size of 25 profiles is also relatively small for a large-scale, generalizable benchmark.

Use of LLMs for evaluations. I know its common practice now but you need to address for bias and reproducibility.

While I appreciate the definition of the personalization metric, this is a complex topic and would help to look for validation in prior research.

**Questions:**

Any details of treatment for bias and reproducibility when running experiments?
How is the "user’s evolving interests, habits, and implicit preferences" incorporated in the profile/evaluation?
Any instances of conflicting signals in the user profiles? How does the DRA handle such cases?
Have you studied the factors that influence the dynamic weight allocation when running eval framework?
What is the cost to run the benchmark? How can we scale it?

---

> ### Author Response · Authors · 2025-11-19
> **Response to Reviewer maAF (W1 & Q5, W2 & Q1)**
>
> Dear reviewer maAF,
>
> Many thanks for your detailed and constructive comments.
>
> ---
>
> **W1 & Q5: Persona authenticity and dataset scale**
>
> We thank the reviewer for raising this valuable point.
>
> 1. Regarding authenticity of personas：
>
> We wish to clarify that our personas are developed based on real-world user donations and subsequent rigorous reconstruction. We initially invited over 50 volunteers to contribute their genuine persona information. Following this donation phase, we performed an intensive data cleaning process to strictly filter out low-quality, incomplete, and repetitive samples. This comprehensive filtering ultimately resulted in 25 high-fidelity user personas, all of which were derived from authentic user contributions and subsequently underwent meticulous privacy de-identification and reconstruction. This methodological approach ensures that our personas are grounded in the complexity of real-world user interests and implicit preferences while maintaining strict privacy standards, thereby maximizing the authenticity of the benchmark.
>
> 2. Benchmark scale (25 profiles × 50 tasks = 250 queries)
>
> For this initial release, our priority was the authenticity and depth of each user profile, including both explicit attributes and dynamic interaction contexts, which required intensive human annotation (5,939 instances; 85 person-days of work and a budget of approximately $6,000 USD), quality control is essential at this pilot stage. Our goal in this initial version was to balance persona diversity, task coverage, and annotation reproducibility:
> - Age range: Teenagers to retirees in their 70s
> - Professional diversity: Engineering, business, humanities, education, arts, medicine, public administration, freelancers, retired professionals
> - Income and socioeconomic variability: Low-income students, middle-class workers, high-income executives
> - Life goals and personal motivations: Academic advancement, career transitions, entrepreneurship, investment, health, parenting
> - Balanced gender representation
>
> Despite only 25 personas, this design allows evaluation across a wide spectrum of personalized deep research behaviors. Due to the computational costs of running complex, multi-step Deep Research Agents (DRAs), most existing deep research benchmarks typically feature no more than 100 queries; however, our dataset, comprising 25 personas with 50 tasks for a total of 250 unique personalized queries, represents a leading scale for this highly complex evaluation paradigm, which provides ample scope for rigorous statistical analysis and has already proven effective in distinguishing system capabilities and revealing critical limitations in current DRAs.
>
> We consider this a pilot-scale benchmark and plan to expand it to include more real profiles across multiple languages and cultures later.
>
> ---
>
> **W2 & Q1: Bias and reproducibility of llm judge**
>
> We sincerely appreciate the reviewer’s thoughtful concern regarding potential bias and reproducibility issues when using LLMs as evaluators. This is an important topic, and we have taken several concrete measures to mitigate bias and improve scoring stability.
>
> In our code implementation, for every scoring criterion within the PQR framework, we mandate the LLM Judge to produce three independent scores. The final score for that criterion is then determined by taking the average of these three scores. This averaging technique effectively smooths out noise and minor fluctuations arising from stochastic model behavior.
>
> To address your concern, we conducted additional robustness testing to empirically examine potential scoring variance. Specifically, we randomly sampled 5 Personalized Deep Research queries, resulting in 300 scoring criteria in total. Each criterion was evaluated 10 times independently using our evaluation code. We analyzed the distribution of scoring fluctuations based on range (max–min) and variance of the 10 repeated scores for each criteria.
>
> | Range | Count | Percent |
> |-------|-------|--------|
> | 0     | 29    | 9.67%  |
> | 1     | 183   | 61.00% |
> | 2     | 81    | 27.00% |
> | 3     | 3     | 1.00%  |
> | 4     | 4     | 1.33%  |
>
> | Variance | Count | Percent |
> |----------|-------|--------|
> | 0 – 0.1  | 71    | 23.67% |
> | 0.1 – 0.2| 44    | 14.67% |
> | 0.2 – 0.3| 111   | 37.00% |
> | 0.3 – 0.4| 25    | 8.33%  |
> | 0.4 – 0.5| 30    | 10%    |
> | 0.5 – 0.6| 6     | 2.00%  |
> | 0.6 – 0.8| 8     | 2.67%  |
> | >0.8     | 5     | 1.67%  |
>
> - Mean score range: 1.23
> - Mean variance: 0.262
>
> These results indicate that the evaluation is stable and reproducible, with the majority of scores exhibiting only minor variation under repeated evaluations. This provides empirical evidence that our scoring system satisfies practical robustness requirements and suggests that LLM-driven assessment does not introduce excessive bias or instability in the benchmark results.

---

> ### Author Response · Authors · 2025-11-19
> **Response to Reviewer maAF (W3, Q2, Q3, Q4)**
>
> Dear reviewer maAF,
>
> Many thanks for your detailed and constructive comments.
>
> ---
>
> **W3: Personalization metric**
>
> We appreciate the reviewer’s suggestion. Our personalization metric is grounded theoretically. PerSE [1] demonstrates the importance of evaluating content according to user-specific preference profiles rather than static global standards. A Survey on Personalized Alignment [2] argues that personalization requires a multi-dimensional evaluation perspective beyond universal correctness. Personality Alignment of LLMs [3] emphasizes that alignment should adapt to individual preferences and decision-making styles rather than only reflecting broad human values. Grounded in these findings, we derived four fundamental dimensions that emerged consistently from our requirement analysis with real users and domain experts:
>
>   1. Goal Alignment: Does the report address explicit and implicit user goals?
>   2. Content Alignment: Suitability of topic, depth, and breadth for user knowledge and interests
>   3. Presentation Fit: Alignment of language, structure, and style
>   4. Actionability & Practicality: Practical value for decision-making
>
> [1] Learning Personalized Alignment for Evaluating Open‑ended Text Generation
> [2] A Survey on Personalized Alignment: The Missing Piece for Large Language Models in Real-World Applications
> [3] Personality Alignment of Large Language Models
>
> ---
>
> **Q2: Incorporating evolving interests, habits, and implicit preferences, about conflicting signals**
>
> In our study, we moved beyond static, structured persona attributes by collecting Dynamic Personalized Contexts (Section 3.2). This was achieved by simulating the daily interactions of our 25 authentic user profiles via a mobile app, recording naturalistic memory snippets (e.g., travel aspirations, health goals) and conversational interactions over time. This longitudinal data is designed to capture each user's evolving interests, habits, and unstated preferences, providing a temporal and behavioral dimension to the profile.
>
> Regarding conflicting signals, we would like to clarify the design principle of our user profiles. In our benchmark construction, the dynamic contexts were meticulously annotated to enrich the user profiles with relevant and reasonable details or "noise" to enhance authenticity (as detailed in our data annotation protocol, Appendix H). The core quality standard was "reversibility," meaning every piece of contextual data must clearly point back to a genuine trait of the user persona. Crucially, annotators were instructed that any added noise "must not contradict other defined attributes in the persona." Therefore, our current benchmark does not contain systematically designed, internally conflicting signals (e.g., a stated budget constraint paired with a preference for luxury goods). The dynamic context is intended to provide complementary, nuanced depth rather than direct contradiction to the structured persona.
>
> ---
>
> **Q3: Factors influencing dynamic weight allocation**
>
> In our PQR framework, weights are dynamically allocated by an LLM meta-evaluator based on the specific Task-Profile pair. We identified prompt design as the critical factor. To ensure robustness, we implemented three strategies:
>
> 1. Iterative Prompt Refinement: We conducted 5 rounds of optimization with manual review to ensure criteria and weights were interpretable, discriminative, and non-overlapping before finalizing the prompt (Sec I.1).
> 2. Ensemble Aggregation: In code implementation, we did not rely on single  llm output. We implemented repeated sampling (5 generations per weight set) followed by aggregation (averaging + normalization + adjustment) to eliminate outliers and ensure stability.
> 3. Model Selection: We experimentally compared GPT-4o, GPT-5, Gemini-2.5-Pro, and Claude-3.7. GPT-5 was selected as the primary meta-evaluator for demonstrating the highest consistency and quality.
>
> ---
>
> **Q4: Cost to run the benchmark**
>
> We use GPT-5 as the judge LLM for the P-Score and Q-Score, and GPT-5-Mini for the R-Score. Based on our empirical measurements, the average cost for evaluating one query is approximately $0.68, covering all three evaluation dimensions. We will clarify the cost more explicitly in the paper.
>
> In addition to evaluation cost, we also measured the cost of generating deep-research report per query across different systems. Results are summarized below:
>
> | Model / Setting | Task Only | Task w/ Persona | Task w/ Context |
> |-----------------|-----------|----------------|----------------|
> | Open-Source DRAs (GPT-5-Mini based) | | | |
> | OAgents | $1.60 | $1.70 | $3.00 |
> | DeerFlow | $0.50 | $0.57 | $1.20 |
> | MiroFlow | $1.00 | $1.11 | $2.10 |
> | LLM-based Agents w/ Search Tools | | | |
> | Gemini-2.5-Pro w/ Search | $0.05 | $0.06 | $0.24 |
> | Claude-3.7-Sonnet w/ Search | $0.03 | $0.04 | $0.31 |
> | Perplexity-Sonar-Reasoning-Pro | $0.02 | $0.03 | $0.78 |
> | GPT-4.1 w/ Search | $0.02 | $0.02 | $0.31 |

---

### Author Response · Authors · 2025-11-21
**Global Response**

Dear all reviewers,

Thank you for your thoughtful reviews!

### **Our Strengths Summarized by Reviewers**

First, we appreciate all of your **positive comments** highlighting the strengths of our work for a summary:

- **Addresses a critical gap through personalization**

  - "The idea here is good. Objective measures of Research Agent outputs are obviously important, but there is value in customizing the research process and the resulting artifact to match the sensibilities of the user. So the work addresses this particular gap." (Reviewer maAF)
  - "The paper focuses on a relatively underexplored aspect of LLM evaluation -- personalization in deep research." (Reviewer vdTb)

- **High-quality, credible, and valuable benchmark**

  - "I expect the benchmark itself would be of some value to the research community." (Reviewer maAF)
  - "In particular the inclusion of structured and context data in user profiles, and the diversity of tasks." (Reviewer maAF)
  - "Real human persona and questions and tasks are designed by human expert that increases the credibility of the benchmark in reflecting real user behaviors." (Reviewer 8bjw)

- **Thorough data collection and evaluation**

  - "The paper collects personalization data in a thorough manner by including people of diverse backgrounds and collect data from these folks over prolonged interactions." (Reviewer vdTb)
  - "The personalization metric is well defined and the evaluations presented are comprehensive." (Reviewer maAF)
  - "Comprehensive analysis across different models and agents." (Reviewer 8bjw)


### **Our main revisions**

We sincerely thank the reviewers for their constructive feedback. In response to the concerns and questions raised, we have mainly conducted the following revisions (A revised draft has been uploaded, with added content marked in blue).

- Clarified theoretical grounding from prior research of the personalization metric (Section 4.1)
- Clarified factuality evaluation methodology and rationale for metric design (Section 4.3)
- Corrected and refined Equation 6, clarified final score computation(Section 4.4)
- Updated evaluation cost details (Section 5.5)
- Add experiment details and cost of report generation across different systems (Section 5.1 and Appendix J)
- Correct the typos

**We will continue to further enhance the quality of this paper according to the discussion with all the reviewers.**

### **Our motivation and contributions**

Last but not least, we wish to **reiterate the motivation and main contributions** of our paper.

#### Motivation
Recent Deep Research Agents (DRAs) show strong potential for complex real-world investigative tasks, yet existing evaluations focus primarily on factual accuracy and report quality, largely overlooking user-specific personalization. In real applications—such as education planning, healthcare decisions, or investment analysis—personalization is essential for actionable value. This gap motivates the need for a benchmark that evaluates DRAs’ ability to adapt reasoning and reporting to diverse user profiles.

#### Contributions

- We formally define the task of personalized deep research, requiring agents to integrate user persona information into retrieval, reasoning, and reporting.
- A benchmark of 250 realistic personalized queries, combining 50 deep research tasks across 10 domains with 25 authentic user profiles.
- A novel evaluation framework jointly measuring:
  - (P) Personalization Alignment
  - (Q) Content Quality
  - (R) Factual Reliability
- Experiments across commercial DRAs, open-source agents, and LLMs with search tools reveal strengths, weaknesses, and open challenges for personalization.

Our dataset is built from authentic user data collection complemented by dynamic contextual interactions, ensuring high realism, diversity, and annotation quality. All tasks and persona pairings undergo committee review to guarantee clarity, complexity, and alignment, providing a rigorous, trustworthy foundation for personalized research evaluation.

### **Thank you!**

We sincerely hope our responses and revisions address all reviewers’ concerns.

We sincerely believe that these updates may help us better deliver the benefits of the proposed work to the ICLR community. Thank you very much!

Authors.

---

### Author Response · Authors · 2025-12-02
**Summary of Rebuttal**

# Summary of Rebuttal

Dear Area Chair,

We would like to express our sincere gratitude for the time you have devoted to our submission under these exceptional circumstances. To help minimize your workload, we respectfully provide a concise summary of our rebuttal:

## **Regarding our rating scores**

We solemnly guarantee strict adhered to the double-blind policy during discussion. We possess no knowledge of reviewer identities and have not used leaked information to manipulate the review process. All interactions were conducted solely based on the scientific content within OpenReview.

We understand that due to the incident, scores have been rolled back to the initial state. However, we respectfully request that you consider the critical consensus reached **before the incident**: Reviewer 8bjw (originally Rating 4) explicitly acknowledged our response and upgraded their score to 6. This implies that the paper had effectively secured **two positive ratings (6 & 6)**. We kindly urge the AC to consider this consensus when evaluating our submission based on the author responses.

The **single** negative review (Reviewer vdTb, rating 2) concerns regarding privacy, evaluation, and metrics. We guaranteed strict data anonymization, demonstrated pipeline stability, and justified our citation-based factuality metric. **We believe these additions have fully addressed the reviewer’s concerns.**

Consequently, we respectfully summarize our rebuttal for your convenience.


## **Regarding reviewers' concerns and our responses**

Please refer to the Global Response and individual Author Responses for full details. We respectfully invite you to consult them at your convenience.

Global Concerns

1. Reliability & Bias of LLM-as-a-Judge (Raised by maAF, 8bjw, vdTb): Reviewers questioned the stability and potential bias of using LLMs for evaluation. We addressed this by:

   - Robustness Testing: Implemented a 3x independent scoring protocol, empirical data showing high stability and low variance (0.262).
   - Expert Alignment: We demonstrated that our LLM judge correlates well with human expert annotations in personalized settings.
   - Prompt Optimization: Detailed our 5-round iterative prompt refinement and ensemble strategy to ensure discriminative criteria.

2. Dataset Scale vs. Cost (Raised by maAF, 8bjw): Concerns were raised regarding the scale (250 queries). We clarified that:
   - Leading Scale: DRAs are computationally expensive, most existing deep research benchmarks typically feature no more than 100 queries. Our 250 unique personalized queries (25 personas × 50 tasks) represent a leading scale.
   - Diversity: We highlighted the rigorous demographic and psychographic diversity of the 25 reconstructed personas.

3. Necessity & Distinction from Existing Benchmarks (Raised by 8bjw): We provided a quantitative correlation analysis showing that PDR performance has low correlation (Spearman ρ ≈ 0.5 or -0.5) with traditional personalization or deep research benchmarks. This empirically proves that PDR evaluates a distinct capability that cannot be approximated by simply combining existing metrics.

Beyond the global issues, we have responded to each reviewer’s individual suggestions as follows.

- maAF (Rating: 6): The reviewer acknowledged the value of the benchmark but asked about persona authenticity and metric theoretical grounding.
  - Response: We detailed the rigorous "donation-cleaning-reconstruction" pipeline for the 25 high-fidelity personas and cited theoretical works (PerSE, Personality Alignment) that ground our PQR metric dimensions.

- Reviewer 8bjw (Rating: 4 $\to$ 6): The reviewer initially questioned the motivation and alignment with user behaviors.
  - Response: We provided a landscape analysis and quantitative evidence of the "Synergistic Necessity" of PDR. We also clarified that our benchmark measures "reasoning correctness" based on persona constraints rather than subjective behavior prediction. The reviewer was satisfied with these clarifications and raised their score.

- Reviewer vdTb (Rating: 2): The reviewer expressed concerns about privacy, factuality metrics, and the λ parameter.
   - Response:
      - Privacy: We confirmed that all data is fully de-identified, abstracted, and compliant with privacy standards, containing zero PII.
      - Factuality: Clarified that our metric measures citation compliance (integrity) rather than atomic fact verification.
      - Technical Details: Corrected Eq. 6 and explained that high P-Scores in "Task w/ Persona" validate the effectiveness of explicit personas over unstructured context.

**We respectfully invite you to examine the full rebuttal and the revised paper**, in which all changes are highlighted in blue for easy reference. **Given that the primary concerns have been fully addressed, we sincerely hope you will find our paper worthy of acceptance.**

Thank you very much again for your invaluable time and consideration!

Best Regards,

Authors

---

### Meta-Review · Program_Chairs · 2026-01-07

**Summary:**

This paper studies the combination of two areas that have become very popular: how to personalize user experiences when using LLMs and how to do deep research effectively. It is one of the first papers tackling this idea, which is important and valuable.

The reviewers were mixed-to-positive; I will summarize all of the concerns as well as the strengths. All reviewers recognized the idea as being valuable and addressing an important limitation. Similarly, most reviewers believed that this benchmark will be valuable to the community. The work performed and the insights already available in the paper are significant as well. Overall, these are very solid strengths.

The larger concerns raised are focused on the reliability of evaluations (for example, given the use of LLM-as-a-judge) and the size of the dataset (is it too small?). One reviewer requested some evidence that personalization is not captured by existing benchmarks (but is by the proposed one). These concerns are all reasonable, and the authors did a good job dealing with them. I felt that the response for handling LLM-as-a-judge bias was solid, the initial dataset size is reasonable given how expensive this type of research can get (and certainly it's large enough to start being useful for community experimentation), and finally the proof of value is good.

Based on this, I think that the paper is valuable and deserves to get in.

**This paper is conditionally accepted provided the authors address the following in the camera ready**
[Ethics concerns] The authors should specify if ethical approval was obtained for the human studies from an ethics review board.

**Reviewer Concerns:**

As noted above, I felt that all of the major concerns are addressed by the rebuttal, so that the paper is in pretty strong shape.

The one concern that could potentially remain is dataset scale---but at the same time, the authors have a reasonably diverse set of personas and a high enough number of queries. Anything substantially larger would require a very large budget, and would not be strictly required for a solid starting point.

Finally, one reviewer was concerned by the potential for privacy violations, but I am not entirely sure why. The authors' protocol for data collection seemed all right to me (informed consent, anonymizing, etc---fairly consistent with work handling personas). Based on the dataset that we end obtaining, the risks appear to be minimal.

**Reviewer Scores:**

R maAF: This reviewer sees the paper as solid to begin with. Many of their questions are easy to clarify (and the authors did so) or have been addressed successfully. I believe the reviewer would have maintained or raised their score.

R 8bjw: This reviewer already had many of their questions answered with a nice rebuttal. The reviewer raised their score and may have further raised it given additional discussion.

R vdTb: This is the more negative reviewer. For the most part, their major concern was around the privacy implications. I felt that the authors did a good job explaining what they do and why this should not be a major issue. All other potential problems were addressed in a solid way. Overall, I think there's a decent chance this reviewer would have raised their score as well.

---

### Decision · Program_Chairs · 2026-01-26

**Decision:**

Accept (Poster)

**Comment:**

Conditions for acceptance have been satisfied.